# PDTrack: Progressive Dense 3D Tracking

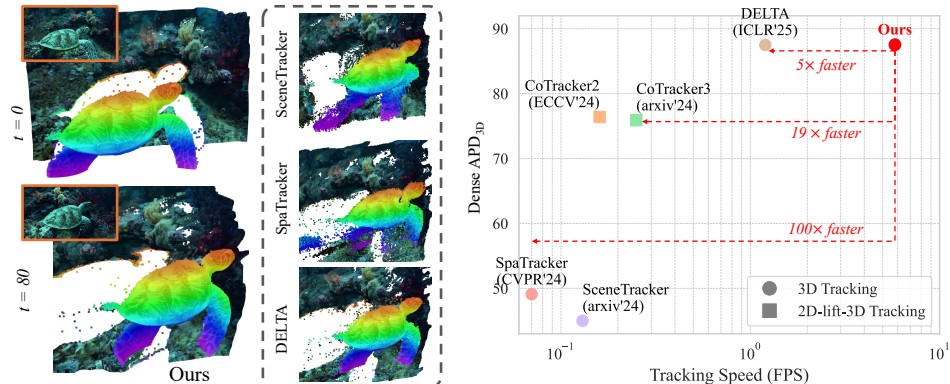

Figure 1: **PDTrack** achieves state-of-the-art for dense 3D tracking — matching the accuracy of DELTA while being $5\times$ faster, and outperforming all prior methods in speed–accuracy tradeoff. (Left) Long-range 3D trajectories on real-world videos. (Right) Performance vs. FPS comparison.

## ABSTRACT

We propose a novel algorithm for accelerating dense long-term 3D point tracking in videos. Through analysis of existing state-of-the-art methods, we identify two major computational bottlenecks. First, transformer-based iterative tracking becomes expensive when handling a large number of trajectories. To address this, we introduce a coarse-to-fine strategy that begins tracking with a small subset of points and progressively expands the set of tracked trajectories. The newly added trajectories are initialized using a learnable interpolation module, which is trained end-to-end alongside the tracking network. Second, we adopt a lightweight implementation for the 4D correlation block that reduces its computational cost on common GPU backends. Together, these improvements lead to a 5–100$\times$ speedup over existing approaches while maintaining state-of-the-art tracking accuracy.

## 1 INTRODUCTION

We focus on *dense 3D tracking*: given an video, we predict the 3D trajectory of every pixel in the first frame across all subsequent frames in the local camera coordinate system. This task is more challenging than related problems such as optical flow (Mémin & Pérez, 1998; Horn & Schunck, 1980; Ilg et al., 2017; Ranjan & Black, 2017; Xu et al., 2017; Sun et al., 2018; Teed & Deng, 2020; Huang et al., 2022b) and scene flow (Quiroga et al., 2014; Teed & Deng, 2021; Liu et al., 2019a; Wang et al., 2020; Niemeyer et al., 2019), which estimate motion only between adjacent frames. In contrast, dense 3D tracking requires maintaining accurate long-term correspondences under larger motions and appearance changes.

To achieve long-term point tracking, recent methods leverage powerful transformer-based networks to estimate trajectories from videos (Harley et al., 2022; Doersch et al., 2023; Li et al., 2024b; Karaev et al., 2023; 2024). Some works (Xiao et al., 2024; Wang et al., 2024a) further extend tracking from 2D to 3D by incorporating accurate monocular depth estimation (Bhat et al., 2023; Piccinelli et al., 2024). However, these approaches remain limited to sparse trajectories, short temporal windows, or both.

DELTA (Ngo et al., 2024) addressed these limitations by introducing the first framework for dense, long-range 3D tracking. While DELTA achieves strong performance and is relatively fast compared to prior work, it remains computationally expensive: tracking every pixel in a 100-frame video takes around 2 minutes, too slow for real-time or latency-sensitive applications.

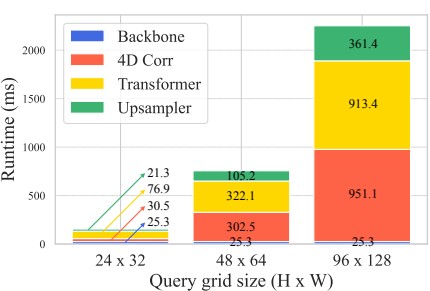

Figure 2: Runtime breakdown of DELTA (Ngo et al., 2024) with a single iteration and one sliding window.

Table 1: Comparison of cost reduction strategies on Kubric3D *val* set (Ngo et al., 2024). All methods use 1 iteration unless noted; full-resolution outputs are obtained via bilinear interpolation.

| Strategy | APD$_{3D}$ $\uparrow$ | Runtime (ms)$\downarrow$ |
|---|---|---|
| DELTA (4 iterations) | 87.3 | 8404 |
| DELTA (1 iteration) | 74.3 | 2275 |
| Downsample video reso. (4×) | 35.8 | **383** |
| Subsample video frames (4×) | 65.4 | 833 |
| Subsample trajectories (16×) | **71.2** | 585 |

In this paper, we present a faster and more scalable successor to DELTA. We begin by identifying two key computational bottlenecks in the DELTA pipeline.

First, despite the efficient transformer design, its iterative refinement must process a large number of trajectory tokens repeatedly for every tracking iteration, leading to high computational cost. We observe that in a dense motion field, many nearby pixels exhibit similar motion patterns. As such, tracking all of them through every iteration is often redundant. Motivated by this, we analyze three possible coarse-to-fine strategies and propose an algorithm that reduces computation by subsampling a sparse set of trajectories in early iterations. The tracking density is then progressively increased in later iterations, culminating in fully dense tracking in the final iteration to ensure accuracy is preserved. To recover the motion of untracked pixels during intermediate iterations, we introduce a learnable interpolation module. It dynamically predicts blending weights to infer the motion of untracked pixels from their nearby tracked neighbors, leveraging both spatial and feature similarity.

Second, through a runtime breakdown for a single iteration of DELTA (see Fig. 2), we observe that 4D correlation feature computation becomes increasingly expensive as the number of tracking points grows, contributing to significant runtime overhead. We identify an inefficiency in the implementation used by prior work that leads to suboptimal GPU utilization and introduce a refinement that improves kernel efficiency without sacrificing performance.

Together, these optimizations achieve a 5× speed-up over DELTA on dense 3D tracking of 100-frame videos, while maintaining state-of-the-art accuracy (Fig. 1).

In summary, our main contribution is bringing dense 3D tracking closer to real-time through the following key design choices: 1) a coarse-to-fine tracking algorithm that starts with sparsely sampled trajectories and densifies them over iterations; 2) a learnable interpolator that produces dense trajectories in each iteration and supports adaptive resampling. 3) an accelerated 4D correlation implementation that improves kernel efficiency on common GPU backends.

## 2 RELATED WORK

**Optical Flow** estimates motion by establishing dense correspondences between consecutive frames. Classical variational methods (Mémin & Pérez, 1998; Horn & Schunck, 1980) faced limitations in handling complex dynamics such as fast motion, occlusions, and large displacements. The advent of CNN-based techniques (Ilg et al., 2017; Ranjan & Black, 2017; Xu et al., 2017) brought substantial improvements to short-term motion estimation. RAFT (Teed & Deng, 2020) introduced a key innovation by computing dense correlation volumes across all pixel pairs. Later approaches expanded on this architecture by introducing transformer-based tokenization of correlation volumes (Huang et al., 2022b), employing global feature aggregation to better address occlusions (Jiang et al., 2021), and modeling optical flow as a softmax-based matching task (Xu et al., 2022). Attempts to extend optical flow to long-term sequences using multi-frame methods (Godet et al., 2021; Shi et al., 2023) or point-tracking integration (Le Moing et al., 2024; Cho et al., 2024a) often face challenges with drift and occlusion, limiting long-term tracking performance.

**Scene Flow** extends optical flow into 3D by estimating dense motion in 3D space. Some methods rely on RGB-D inputs (Hadfield & Bowden, 2011; Hornacek et al., 2014; Quiroga et al., 2014;

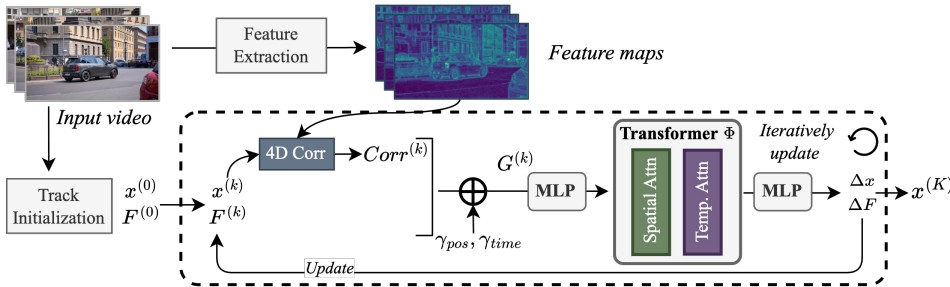

Figure 3: A modern long-range tracking pipeline (we omit the 3D tracking parts here for simplicity)

Teed & Deng, 2021; Yang & Ramanan, 2021), while others operate on monocular video alone (Hur & Roth, 2020; 2021; Bayramli et al., 2023; Jiang & Okutomi, 2023). A separate line of work estimates 3D motion directly from point clouds (Liu et al., 2019a; Gu et al., 2019; Niemeyer et al., 2019). Several approaches (He et al., 2022; Liu et al., 2019b; Huang et al., 2022a) leverage temporal context to predict scene flow from videos. More recently, methods such as (Sucar et al., 2025; Liang et al., 2025) build on top of foundation 3D reconstruction models (Wang et al., 2024c; Leroy et al., 2024) to estimate 3D scene flow in a more scalable manner.

**Point Tracking** estimates long-range motion trajectories in videos. Particle Video (Sand & Teller, 2008) introduced particle trajectories for long-range video motion. TAP-Vid (Doersch et al., 2022) provided a comprehensive benchmark to evaluate point tracking and TAPNet, a baseline that predicts tracking locations using global cost volume and soft-argmax operation. PIPs (Harley et al., 2022) revisited the concept of particle video and proposed a network that updates trajectories iteratively over fixed temporal windows, but ignored spatial context with independent point tracking and struggle with occlusion. Subsequent efforts further improve tracking performance by jointly process multiple points with temporal attention and spatial attention (Karaev et al., 2023; 2024); using 4D correlation feature (Cho et al., 2024b). SceneTracker (Wang et al., 2024a) and SpaTracker (Xiao et al., 2024) extend point tracking to 3D by incorporating depth information, but remain inefficient for dense tracking due to computationally expensive cross-track attention. DELTA (Ngo et al., 2024) is the first approach tackling the dense 3D tracking problem with an efficient transformer and upsample layer to obtain high-resolution dense tracking. **Concurrently**, several other efforts also explore 3D tracking. Seurat (Cho et al., 2025) estimates depth changes on top of a 2D tracker (Karaev et al., 2023) to recover 3D motion. TAPIP3D (Zhang et al., 2025) proposes a 3D-space correlation mechanism for directly tracking points in 3D space. St4RTrack (Feng et al., 2025) extends 3D reconstruction models (Wang et al., 2024c) for dense 3D tracking via joint optimization.

**Tracking by Reconstructing** tackles long-range motion estimation by explicitly reconstructing a deformable scene representation. OmniMotion (Wang et al., 2023b) jointly optimizes a NeRF (Mildenhall et al., 2020) with a bijective deformation field (Dinh et al., 2016) and extracts 2D point trajectories through this mapping. More recent efforts leverage DINOv2 (Oquab et al., 2023) to compute long-range correspondences, either through enhanced invertible deformation fields (Song et al., 2024) or self-supervised techniques (Tumanyan et al., 2024). Shape-of-Motion (Wang et al., 2024b) uses 3D Gaussian Splatting to jointly learn geometry and motion, enabling point tracking by tracing Gaussian positions across frames. While these reconstruction-based methods are capable of producing dense trajectories, they rely on per-video optimization, making them significantly slower and less accurate than feedforward, data-driven tracking models on standard benchmarks.

## 3 METHOD

**Problem setup:** The input to our method is an RGB-D video, where the RGB frames are represented as $V \in \mathbb{R}^{T \times H \times W \times 3}$, with $T$, $H$, and $W$ denoting the temporal and spatial dimensions. The corresponding depth maps $D \in \mathbb{R}^{T \times H \times W}$ are generated using a monocular depth estimation model. Our model outputs dense, occlusion-aware 3D trajectories $P \in \mathbb{R}^{T \times H \times W \times 4}$, where each vector $\mathbf{p}_{(t,u,v)} = (u_t, v_t, d_t, o_t)$ captures the motion of a pixel located at $(u, v)$ in the first frame as it propagates to frame $t$. Here, $(u_t, v_t)$ denote the projected 2D pixel coordinates in frame $t$, $d_t$ is the predicted depth, and $o_t \in \{0, 1\}$ indicates the visibility status.

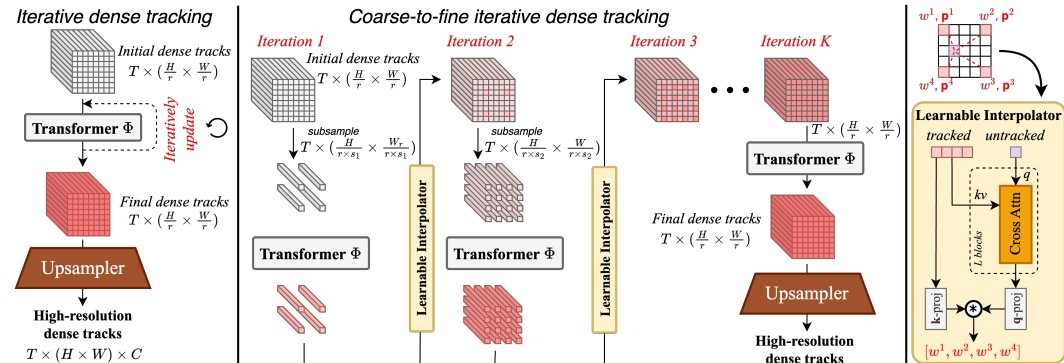

Figure 4: **Overview of our proposed framework**. (Left) Traditional iterative dense tracking refines all trajectories at every iteration, leading to high computational cost. (Middle) Our **coarse-to-fine iterative dense tracking** reduces computation by subsampling trajectory points in early iterations and progressively increasing the density across iterations. (Right) A **learnable interpolation module** leverages attention to infer untracked motions from nearby tracked pixels, enabling efficient and adaptive trajectory propagation.

### 3.1 PRELIMINARY: MODERN TRACKING FRAMEWORKS

Our work builds upon a modern tracking framework. We select DELTA (Ngo et al., 2024), a recent approach for dense 3D tracking from RGB-D videos, but other frameworks have a similar structure (See Fig. 3 for an overview). First, a feature extraction network $F$ (similar to Teed & Deng (2020)) creates image feature map for each frame. Then, the core of the algorithm iteratively evolves a data structure of size of dense tracks $\mathcal{D} \in \mathbb{R}^{T \times \frac{H}{r} \times \frac{W}{r}}$, denoted as $\{P_i\}$, where $i$ is the trajectory index, and $P_i = [\boldsymbol{p}_1^i, \boldsymbol{p}_2^i, \cdots, \boldsymbol{p}_T^i]$, with $\boldsymbol{p}_t^i = (u_t^i, v_t^i, d_t^i, o_t^i)$ being the 3D location and visibility of the point associated with the $i$-th trajectory at the $t$-th frame. The downsampling factor $r$ is beneficial for faster processing.

At each iteration, a trajectory is represented by a list of tokens $G^i = [G_1^i, G_2^i, \cdots, G_T^i]$, each token $G_t^i$ encodes position, visibility, appearance and correlation of the trajectory at $t$-th frame:

$$G_t^i = [\mathcal{F}_t^i, Corr_t^i, DCorr_t^i, o_t^i, \gamma(\boldsymbol{x}_t^i - \boldsymbol{x}_1^i)] + \gamma_{pos}(\boldsymbol{x}_t^i) + \gamma_{time}(t), \tag{1}$$

where $\mathcal{F}_t^i$ are image features extracted by $F$, $Corr_t^i$ are correlation features (Teed & Deng, 2020; Karaev et al., 2023; Cho et al., 2024b), $DCorr_t^m$ is depth correlation, and $\gamma_{pos}$ and $\gamma_{time}$ are the positional embedding of the input position $\boldsymbol{x}_t^i = (u_t^i, v_t^i, d_t^i)$ and time $t$, respectively. For more details about the individual components, refer to Ngo et al. (2024).

The tokens $G_t^i$ are processed by a transformer network $\Phi$. From the output tokens of $\Phi$ we can compute new dense tracks using a pointwise MLP and use the computed point tracks and the output tokens to compute the input tokens for the next iteration. The framework uses $K$ iterations, e.g., $K = 4$. The network $\Phi$ itself employs a joint global-local spatial attention mechanism: global attention over a sparse set of anchor tracks captures long-range motion, while local attention focuses on fine-grained motion within small spatial neighborhoods.

Finally, a transformer-based upsampling module is used to upsample the dense tracks to the full resolution $T \times H \times W \times 4$. This upsampler models each pixel's trajectory as a soft combination of neighboring trajectories (see Ngo et al. (2024) for details of this architecture).

While DELTA achieves end-to-end dense tracking and has been optimized for runtime, we undertake the challenging task of finding further significant improvements. We will discuss this next.

### 3.2 ANALYSIS OF STRATEGIES FOR ACCELERATING TRACKING

To accelerate the tracking framework, we first profile the time usage of each stage in the tracking algorithm, as shown in Fig. 2. This analysis reveals two primary bottlenecks: (1) the point tracking transformer $\Phi$, and (2) the computation of the correlation features $Corr$. In this section, we focus on analyzing of the transformer $\Phi$.

The computational cost of $\Phi$ scales linearly with the number of trajectories (Karaev et al., 2023), and quadratically with the number of frames. Therefore, reducing either the number of trajectories or the number of frames can lead to substantial speedups. We consider three factors that influence these quantities: the spatial resolution of the input RGB images, the temporal resolution of the video, and the number of trajectory points tracked. To understand the impact of these factors on both performance and efficiency, we design a series of tests to evaluate one iteration of tracking. For each setting, we evaluate two aspects: (1) the time required for computation, and (2) the degradation in tracking accuracy, measured by the $APD_{3D}$ metric on a synthetic dataset. The specific tests are described below.

**1) Spatial downsampling of input RGB images:** We reduce the resolution of the input frames by downsampling both the width and height by a factor of 4. Point tracking is then performed on the low-resolution frames using the pretrained transformer $\Phi$. We then upsample the resulting trajectories back to the original resolution for evaluation. This analysis reduces both the number of trajectories and the spatial cost of attention, providing insights into how resolution affects accuracy and speed.

**2) Temporal subsampling of input frames:** We subsample the input video along the temporal dimension by a factor of 4, reducing the number of frames used in trajectory computation. The trajectories are still computed at full spatial resolution. Afterward, we apply linear interpolation to upsample the trajectories back to match the original temporal length of the video. Although more advanced interpolation methods such as B-splines or non-linear motion kernels could be used to model more complex dynamics, linear interpolation is sufficient for analyzing the effects of temporal subsampling on performance.

**3) Subsampling the number of trajectories:** We keep the input frames at full spatial and temporal resolution, but subsample the initial set of trajectory points by a factor of 16. This reduces the number of tokens passed through the transformer while preserving the original feature resolution.

4) **Baseline:** no downsampling is used as reference. We denote this as DELTA (1 iteration) as reference and DELTA (4 iterations) for the complete algorithm.

We present the results of this analysis in Tab. 1 (see the *supplementary* for more details). Among the evaluated strategies, *subsampling the number of trajectories* emerges as the most effective, as other approaches incur significantly higher tracking errors with clear drawbacks. *Spatial downsampling* fails due to information loss caused by reduced feature map resolution. *Temporal subsampling* assumes linear motion over time, but DELTA (Ngo et al., 2024) tracks in UVD space, where 3D linear motion does not correspond to linear 2D and depth trajectories. As a result, interpolating skipped frames introduces distortion and lowers accuracy. Based on this analysis, we propose a new algorithm that leverages trajectory subsampling as a core component (see Sec. 3.3.1). To complement this, we introduce a novel learnable interpolation architecture to recover dense trajectories, forming a key component of our method (see Sec. 3.3.2). Finally, we present an accelerated design for computing correlation features, further improving overall efficiency (see Sec. 3.3.3).

In addition to the three major contributions discussed above, we explored several alternative design choices to further accelerate the tracking framework. These include reducing the number of transformer layers (e.g., from 6 to 3, see Tab. 8), lowering the number of iterative refinements (see Fig. 5), and experimenting with various trajectory subsampling strategies (see Tab. 5).

## 3.3 ACCELERATING DENSE POINT TRACKING

### 3.3.1 COARSE TO FINE TRACKING WITH TRAJECTORY SUBSAMPLING

As shown in Tab. 1, tracking on a subsampled set of points followed by interpolation significantly reduces runtime while maintaining accuracy for the interpolated points. This observation naturally leads to a coarse-to-fine tracking algorithm. We begin by sampling a sparse set of points and perform one iteration of point tracking. Next, we interpolate the positions of the remaining untracked points. These interpolated positions provide reasonable estimates that serve as initialization for a denser set of points in the next iteration. The process is repeated, using the interpolated estimates to guide the position updates for increasingly dense sets of points, progressively refining the tracking results.

More concretely, the algorithm proceeds as follows.

**1) Subsample points.** We begin by subsampling a small set of points to track from a $\frac{H}{r} \times \frac{W}{r}$ pixel grid, where $r$ is the upsampling ratio of the upsampler used in DELTA. In our implementation (illustrated in Fig. 4), we uniformly sample points on a sparse grid of size $\frac{H}{r \times s_1} \times \frac{W}{r \times s_1}$, where $s_1$ denotes the initial spatial subsampling scale for the first iteration.

Each sampled point is initialized with its original pixel position. After each coarse-to-fine tracking iteration, we update the positions of the points that have already been tracked and introduce additional points for tracking in the next iteration. At the $k$-th iteration, the number of points tracked corresponds to a grid of size $\frac{H}{r \times s_k} \times \frac{W}{r \times s_k}$, where $s_k$ is the subsampling scale at step $k$. In our implementation, we decrease $s_k$ by a factor of 2 at each step, progressively refining the tracking resolution.

In our ablation study, we compare different subsampling schedules and explore alternative strategies, such as random sampling and importance sampling focused on distinctive features (e.g., object boundaries). However, we find that the simple uniform grid sampling performs on par with these more complex strategies. Therefore, we adopt the grid-based approach for its simplicity.

**2) Single-step point tracking.** We compute the correlation features for each sampled point based on its current estimated or initialized position, and feed these features into the transformer $\Phi$ to produce updated point positions.

**3) Interpolate positions for new points.** Given the current estimated positions of the sampled points, we use an interpolation method to estimate the initial positions of the additional points introduced at the next scale, which have not yet been tracked. We experimented with various interpolation strategies, including bilinear and nearest-neighbor interpolation, and found that nearest-neighbor consistently outperforms bilinear (see Fig. 5). We hypothesize that bilinear interpolation tends to oversmooth positional estimates, especially in regions with large motion differences, which is a problem that becomes more pronounced in long-term tracking scenarios. To further improve the accuracy of interpolated positions, we introduce a learnable interpolation module (see Sec. 3.3.2), which is trained end-to-end alongside the coarse-to-fine tracking pipeline.

The algorithm then returns to step **1)** and repeats the process until the full resolution is reached.

It is noteworthy that our scheme is *different* and *orthogonal* to DELTA's. DELTA performs a *frame-level* coarse step—downsampling once yet still tracking *all* low-resolution pixels at every refinement—so the per-iteration attention cost remains unchanged. We instead adopt a *token-level* schedule: start with a very sparse subset, initialize newly activated tokens via a feature-aware interpolator, progressively densify across iterations, and run a single final full-grid pass. This removes the heavy attention cost from early iterations while preserving pixel-level accuracy.

### 3.3.2 Learnable Interpolation Module

To propagate motion from tracked to untracked pixels during coarse-to-fine iterations, we introduce a *learnable interpolation module* that estimates the 3D motion of each untracked pixel as an adaptive blend of nearby tracked motions. Unlike fixed interpolation methods (e.g., bilinear or nearest-neighbor), our approach dynamically predicts interpolation weights using attention over spatial features. Let $\mathcal{P}_{\text{track}}$ be the set of tracked pixels and $\mathcal{P}_{\text{query}}$ the untracked ones. For each query $(u, v) \in \mathcal{P}_{\text{query}}$, we predefine its four nearest neighbors $\{(u_j, v_j)\}_{j=1}^4 \in \mathcal{P}_{\text{track}}$. The interpolated 3D motion $\mathbf{p}_{(u,v)}$ (for clarity, we omit the frame subscript $t$) is computed as:

$$\mathbf{p}_{(u,v)} = \sum_{j=1}^{4} w_{(u,v)}^j \cdot \mathbf{p}_{(u_j, v_j)} \qquad (2)$$

where weights $w_{(u,v)}^j \in [0, 1]$ sum to 1 and are predicted by a lightweight attention module.

Let $\mathcal{F} \in \mathbb{R}^{\frac{H}{r} \times \frac{W}{r} \times C_{\mathcal{F}}}$ denote the dense feature map of the query frame of the dense tracking from the feature extractor $F$. The initial query feature $\mathcal{F}_{(u,v)}$ is first refined via $L$ multi-head cross-attention blocks over the support set. We follow the Alibi attention scheme (Press et al., 2022) in these blocks, where a relative positional bias is added to the attention logits, computed as the L1 distance between the query and support pixel locations. The refined query and the final weights are predicted as:

$$\tilde{\mathcal{F}}_{(u,v)} = \text{CrossAttn}^{(L)}\left(\mathcal{F}_{(u,v)}, \{\mathcal{F}_{(u_j, v_j)}\}\right), \mathbf{w}_{(u,v)} = \text{softmax}\left(\boldsymbol{q}(\tilde{\mathcal{F}}_{(u,v)}) \cdot \boldsymbol{k}\left(\{\mathcal{F}_{(u_j, v_j)}\}\right)\right) \quad (3)$$

Table 2: **Long-range optical flow results** on CVO (Wu et al., 2023; Le Moing et al., 2024).

| Methods | CVO-Clean (7 frames) | | CVO-Final (7 frames) | | CVO-Extend (48 frames) | |
|---|---|---|---|---|---|---|
| | EPE↓ (*all/vis/occ*) | IoU↑ | EPE ↓ (*all/vis/occ*) | IoU↑ | EPE↓ (*all/vis/occ*) | IoU↑ |
| RAFT | 2.48 / 1.40 / 7.42 | 57.6 | 2.63 / 1.57 / 7.50 | 56.7 | 21.80 / 15.4 / 33.4 | 65.0 |
| MFT | 2.91 / 1.39 / 9.93 | 19.4 | 3.16 / 1.56 / 10.3 | 19.5 | 21.40 / 9.20 / 41.8 | 37.6 |
| TAPIR | 3.80 / 1.49 / 14.7 | 73.5 | 4.19 / 1.86 / 15.3 | 72.4 | 19.8 / 4.74 / 42.5 | 68.4 |
| CoTracker2 | 1.51 / 0.88 / 4.57 | 75.5 | 1.52 / 0.93 / 4.38 | 75.3 | 5.20 / 3.84 / 7.70 | 70.4 |
| DOT | 1.29 / 0.72 / 4.03 | **80.4** | 1.34 / 0.80 / 3.99 | **80.4** | 4.98 / 3.59 / 7.17 | **71.1** |
| SceneTracker | 4.40 / 3.44 / 9.47 | - | 4.61 / 3.70 / 9.62 | - | 11.5 / 8.49 / 17.0 | - |
| SpatialTracker | 1.84 / 1.32 / 4.72 | 68.5 | 1.88 / 1.37 / 4.68 | 68.1 | 5.53 / 4.18 / 8.68 | 66.6 |
| DOT-3D | 1.33 / 0.75 / 4.16 | 79.0 | 1.38 / 0.83 / 4.10 | 78.8 | 5.20 / 3.58 / 7.95 | 70.9 |
| DELTA | **0.94 / 0.51 / 2.97** | 78.7 | **1.03 / 0.61 / 3.03** | 78.3 | 3.67 / 2.64 / 5.30 | 70.1 |
| Ours | 1.04 / 0.61 / 3.25 | 77.6 | 1.12 / 0.69 / 3.27 | 77.3 | **3.53 / 2.57 / 5.10** | 70.6 |

where $q(\cdot)$ and $k(\cdot)$ are linear projections for the refined query and support features. This mechanism allows sparse trajectories to efficiently guide the evolution of the full dense motion field over time, maintaining temporal/spatial consistency without the cost of processing full dense trajectories.

### 3.3.3 ACCELERATING COMPUTATION OF CORRELATION FEATURES

A major bottleneck in DELTA (Ngo et al., 2024) is the computation of 4D correlation features, introduced in Cho et al. (2024b), which measure pairwise similarities between local neighborhoods around query and predicted positions. These features form a 4D tensor processed by a dual-convolutional module. As shown in our profiling (Fig. 2), this becomes increasingly expensive with a large number of trajectories.

The main inefficiency arises from the input channel size (typically $7 \times 7 = 49$), which is not divisible by 8, leading to poor GPU utilization in dense settings. To address this, we add a lightweight MLP projection that reduces the dimension to 32, followed by LayerNorm and ReLU. This adds minimal overhead for sparse tracking and yields a significant speedup in the dense setting. An alternative from Karaev et al. (2024) replaces the dual-conv module with pure MLPs on flattened 4D correlation, but we find it performs worse on dense 2D/3D tracking (See Tab. 8).

## 4 EXPERIMENTS

**Implementation Details:** Following prior work (Ngo et al., 2024; Karaev et al., 2023; Xiao et al., 2024), we use the Kubric engine (Greff et al., 2022) to generate 5,631 RGB-D training videos with both sparse and dense tracking annotations. We supervise the model on both sparse and dense tracks. The total loss combines the 2D coordinate loss $\mathcal{L}_{2D}$, inverse depth loss $\mathcal{L}_{depth}$, and visibility loss $\mathcal{L}_{visib}$. The loss weights $\lambda_{2d}, \lambda_{depth}, \lambda_{visib}$ follow those in Ngo et al. (2024). We initialize the model from the pretrained DELTA checkpoint and add parameters for the interpolation module. The model is trained for 100,000 iterations on 8 A100 GPUs using AdamW with a one-cycle scheduler, starting from a learning rate of $2 \times 10^{-4}$.

### 4.1 COMPARISON WITH PRIOR WORK

**Baselines.** We compare our method with previous optical flow and point tracking approaches. In particular, we closely compare it with DELTA (Ngo et al., 2024), the current SOTA for dense 3D tracking. We also include DOT (Le Moing et al., 2024), a method for dense 2D tracking, its 3D variant DOT-3D (Ngo et al., 2024), and other recent point tracking methods, including Co-Tracker2 (Karaev et al., 2023), CoTracker3 (Karaev et al., 2024), SpaTracker (Xiao et al., 2024), and SceneTracker (Wang et al., 2024a).

**Benchmark Datasets.** We evaluate our method on a diverse set of tracking benchmarks covering 2D optical flow, dense 3D pixel tracking, and sparse 3D point tracking. (1) Long-range 2D optical flow: We use the **CVO** dataset (Wu et al., 2023; Le Moing et al., 2024), which includes three splits: *CVO-Clean*, *CVO-Final*, and *CVO-Extended*. Each split contains about 500 videos with 7 frames (48 frames for *CVO-Extended*), captured at 60 FPS, with dense long-range optical flow annotations

Table 3: **Dense 3D tracking results** on the Kubric3D dataset. The runtimes exclude the time for depth estimation, which is precomputed for all methods using the same settings.

| Methods | Kubric-3D (24 frames) | | | Runtime↓ |
| --- | --- | --- | --- | --- |
| | AJ↑ (*all/vis*) | APD$_{3D}$ ↑ (*all/vis*) | OA↑ | (second) |
| CoTracker2 | 70.0 / 80.7 | 76.4 / 85.1 | 96.7 | 145 |
| CoTracker3 | 68.9 / 79.3 | 75.9 / 84.3 | 95.8 | 94 |
| SpatialTracker | 35.3 / 42.7 | 49.1 / 51.6 | 96.5 | 350 |
| SceneTracker | - | 45.0 / 65.5 | - | 179 |
| DOT-3D | 68.1 / 72.3 | 75.3 / 77.5 | 88.7 | 11.8 |
| DELTA | **81.7** / **85.1** | 87.5 / 90.9 | **97.5** | 18.2 |
| Ours | **81.7** / 84.8 | **87.6** / **91.0** | 97.0 | **3.5** |

Table 4: **3D tracking results** on the TAP-Vid3D Benchmark. We use Unidepth-V2 for depth estimation.[†] denotes using depth to lift 2D tracks to 3D tracks.

| Methods | Aria | | | DriveTrack | | | PStudio | | | Average | | |
| --- | --- | --- | --- | --- | --- | --- | --- | --- | --- | --- | --- | --- |
| | AJ↑ | APD$_{3D}$ ↑ | OA↑ | AJ↑ | APD$_{3D}$ ↑ | OA↑ | AJ↑ | APD$_{3D}$ ↑ | OA↑ | AJ↑ | APD$_{3D}$ ↑ | OA↑ |
| TAPIR[†] + COLMAP | 7.1 | 11.9 | 72.6 | 8.9 | 14.7 | 80.4 | 6.1 | 10.7 | 75.2 | 7.4 | 12.4 | 76.1 |
| CoTracker2[†] + COLMAP | 8.0 | 12.3 | 78.6 | 11.7 | 19.1 | 81.7 | 8.1 | 13.5 | 77.2 | 9.3 | 15.0 | 79.1 |
| BootsTAPIR[†] + COLMAP | 9.1 | 14.5 | 78.6 | 11.8 | 18.6 | 83.8 | 6.9 | 11.6 | 81.8 | 9.3 | 14.9 | 81.4 |
| CoTracker2[†] + UniDepth | 13.0 | 20.9 | 84.9 | 12.5 | 19.9 | 80.1 | 6.2 | 13.5 | 67.8 | 10.6 | 18.1 | 77.6 |
| TAPTR[†] + UniDepth | 15.7 | 24.2 | 87.8 | 12.4 | 19.1 | 84.8 | 7.3 | 13.5 | **84.3** | 11.8 | 18.9 | **85.6** |
| LocoTrack[†] + UniDepth | 15.1 | 24.0 | 83.5 | 13.0 | 19.8 | 82.8 | 7.2 | 13.1 | 80.1 | 11.8 | 19.0 | 82.3 |
| SpatialTracker + UniDepth | 13.6 | 20.9 | **90.5** | 8.3 | 14.5 | 82.8 | 8.0 | **15.0** | 75.8 | 10.0 | 16.8 | 83.0 |
| SceneTracker + UniDepth | - | 23.1 | - | - | 6.8 | - | - | 12.7 | - | - | 14.2 | - |
| DOT-3D + UniDepth | 13.8 | 22.1 | 85.5 | 11.8 | 17.9 | 82.3 | 3.2 | 5.3 | 52.5 | 9.6 | 15.1 | 73.4 |
| DELTA + UniDepth | 16.6 | 24.4 | 86.8 | 14.6 | 22.5 | **85.8** | **8.2** | **15.0** | 76.4 | 13.1 | 20.6 | 83.0 |
| Ours + UniDepth | **17.0** | **24.7** | 87.2 | **15.6** | **23.8** | 84.6 | 7.7 | 14.4 | 74.1 | **13.4** | **21.0** | 81.2 |

and occlusion masks. (2) Dense 3D pixel tracking: Following Ngo et al. (2024), we evaluate on a held-out test split of the **Kubric** dataset (Greff et al., 2022), which includes 143 RGB-D videos, each 24 frames long, with dense ground-truth 3D trajectories for every pixel. (3) 3D point tracking: We benchmark on the large-scale **TAP-Vid3D** dataset (Koppula et al., 2024), which contains 4,569 videos from DriveTrack (Balasingam et al., 2024), PStudio (Joo et al., 2017), and Aria (Pan et al., 2023), covering a range of real-world and simulated environments, with video lengths from 25 to 300 frames.

**Metrics.** Please see our appendix for the details of evaluation metrics.

**Long-range 2D optical flow.** Our method achieves comparable performance to DELTA (Ngo et al., 2024), a state-of-the-art approach for dense tracking (see Tab. 2). While we observe slightly lower accuracy on the *Clean* and *Final* subsets, our model outperforms DELTA on the more challenging *Extended* subset, which contains significantly longer video sequences. This demonstrates our model's robustness in long-range tracking scenarios.

**Dense 3D Tracking.** Table 3 reports results on the Kubric3D test set. Our method matches the performance of DELTA while significantly outperforming other baselines, including 3D tracking methods such as SpaTracker (Xiao et al., 2024) and SceneTracker (Wang et al., 2024a), as well as 2D-to-3D lifted approaches like CoTracker2 (Karaev et al., 2023) and CoTracker3 (Karaev et al., 2024). Importantly, our approach is substantially more efficient: it achieves up to a $5\times$ speed-up over DELTA and is approximately $100\times$ faster than SPATRACKER, enabling practical deployment in real-time or large-scale applications. Qualitative results of dense 3D tracking on in-the-wild videos are provided in the *supplementary materials*.

**3D Tracking.** We evaluate our approach on sparse 3D point tracking in Table 4. Although the sparse setting does not directly benefit from our coarse-to-fine strategy, the results demonstrate that training with this strategy—and incorporating our redesigned 4D correlation module—does not degrade sparse tracking performance. Our method achieves slightly better results on the *Aria* and *DriveTrack* subsets, with a minor drop on *PStudio*, showing overall competitive and robust performance.

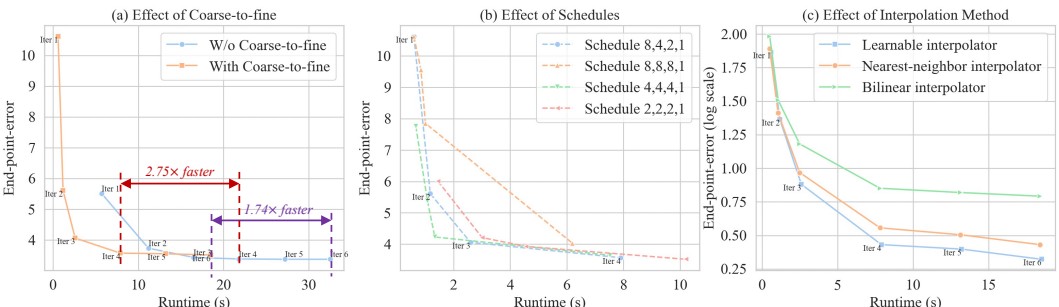

Figure 5: **Analysis of the coarse-to-fine strategy.** We visualize how accuracy evolves with runtime across different methods. (a) We evaluate runtime per iteration and observe that the coarse-to-fine strategy consistently reduces runtime compared to the baseline while achieving similar accuracy. (b) We compare different coarse-to-fine scheduling strategies. (c) Nearest neighbor interpolation outperforms bilinear, and our learnable interpolation further improves accuracy.

Table 5: Ablation of sampling strategies.

| Sampling | EPE↓ | Runtime (s) |
|---|---|---|
| None | 3.50 / 2.55 / 5.09 | 21.8 |
| Uniform grid (default) | 3.53 / 2.57 / 5.10 | 7.6 |
| Random | 3.51 / 2.61 / 5.04 | 7.8 |
| SIFT | 3.79 / 2.88 / 5.72 | 7.8 |

Table 6: Impact of number of neighbor.

| #. of neighbors | EPE↓ |
|---|---|
| 2 | 3.57 / 2.60 / 5.19 |
| 4 (default) | 3.53 / 2.57 / 5.10 |
| 8 | 3.55 / 2.58 / 5.16 |
| 16 | 3.60 / 2.63 / 5.27 |

## 4.2 ABLATION STUDY

We conduct a series of ablation studies on the CVO *Extended* split (Le Moing et al., 2024). Runtime is measured as the average time required to densely track all $384 \times 512$ pixels across a 48-frame video using a machine with a single A100 GPU. The number of iterations is set to 4 by default, unless otherwise specified.

**Analysis of the Coarse-to-fine strategy.** Figure 5a compares our coarse-to-fine strategy (schedule: 8,4,2,1) with the baseline that tracks all trajectories in every iteration. Our method achieves similar accuracy while being $2.75\times$ faster. Fig. 5b compares different subsampling schedules, where $(s_1, s_2, s_3, s_4)$ denotes the subsampling factors across 4 iterations. A finer schedule (2,2,2,1) improves accuracy but with increased runtime. In Fig.5c, we ablate different interpolators, showing that our attention-based design outperforms fixed alternatives like nearest-neighbor and bilinear.

Table 5 presents different pixel sampling strategies. In addition to the default uniform grid sampling, we evaluate two alternatives: random sampling and SIFT-based keypoint selection. Both alternatives require an additional step to identify neighbors for the interpolation module. While the random strategy yields slightly better results, the SIFT-based approach performs worse, likely due to non-uniform coverage. We retain the uniform grid sampling as our default due to its simplicity, efficiency, and competitive performance.

We ablated the interpolator with $J \in \{2, 4, 8, 16\}$ neighbors in Table 6. Using too few neighbors ($J = 2$) hurts performance, while using too many ($J = 16$) also worsens results by averaging over

Table 7: Latency of 4D correlation on different devices.

| Device | DELTA's conv. | Our conv. | Speed-up |
|---|---|---|---|
| A100 80Gb | 162.5 | 44.2 | 3.6× |
| RTX4090 24Gb | 103.3 | 30.9 | 3.3× |
| T4 16Gb | 529 | 150 | 3.5× |

Table 8: Impact of components on runtime.

| Model Variant | EPE↓ | Runtime (s) |
|---|---|---|
| DELTA (Ngo et al., 2024) | 3.67 / 2.64 / 5.30 | 32.1 |
| + new 4D Corr | 3.50 / 2.55 / 5.09 | 21.8 |
| + Coarse-to-fine | 3.53 / 2.57 / 5.10 | 7.6 |
| + 3-layer Trans | 3.91 / 2.85 / 5.82 | 5.8 |
| + MLP 4D Corr | 4.76 / 2.81 / 7.75 | 5.4 |

Table 9: Results of CoTracker3 (Karaev et al., 2024) for dense 2D tracking.

| Setting | Clean EPE (all/vis/occ) | Final EPE (all/vis/occ) | Extended EPE (all/vis/occ) | Time |
|---|---|---|---|---|
| W/o Coarse-to-fine | 1.19 / 0.72 / 3.78 | 1.21 / 0.75 / 3.72 | 4.28 / 2.63 / 7.03 | 10.1 s |
| With Coarse-to-fine | 1.21 / 0.71 / 4.03 | 1.32 / 0.82 / 4.01 | 4.37 / 2.73 / 7.27 | 3.7 s |

irrelevant supports that may not share the same motion pattern and thus over-smooth boundaries. Therefore, we use $J = 4$ by default as the best trade-off.

**Ablation of the 4D correlation.** In Table 7, we measured the wall-clock time of the dual-conv module of a single 4D correlation call (identical input: a window of 16 frames and resolution of $384 \times 512$, batch= 1) on several devices. "DELTA" is the original implementation; "Our" is the projected 32-channel version proposed in this paper. Across hardware, our variant consistently reduces the latency by roughly $3\times$ (often more), yielding a substantial speedup of this module.

**Runtime Analysis.** Table 8 summarizes our runtime improvements. Replacing the original 4D correlation in DELTA reduces runtime by 33%. Adding our coarse-to-fine strategy yields a further $3\times$ speed-up. A 3-layer transformer brings runtime to 5.8s, though with some loss in accuracy.

**Ablation on different tracking baseline.** Our coarse-to-fine strategy is not tied to a specific base tracker. We applied our method to CoTracker3 (Karaev et al., 2024) for dense 2D tracking to demonstrate its generalizability. As shown in Tab. 9, integrating our coarse-to-fine pipeline with CoTracker3 yields a $3\times$ speed-up with only marginal accuracy degradation.

## 5 CONCLUSION

In this work, we presented an efficient framework for dense 3D video tracking that significantly improves the runtime of DELTA (Ngo et al., 2024) while preserving its strong performance. We proposed a coarse-to-fine tracking algorithm that progressively increases spatial coverage across iterations, combined with a learnable interpolation module for dense supervision and a faster 4D correlation computation. These improvements yield $5 - 100\times$ speedup over previous approaches, making our approach more suitable for real-time applications. Nonetheless, our method inherits common **limitations** of data-driven tracking: it is trained on synthetic data and may struggle under fast motion, severe occlusions, or poor depth estimation, which affect 2D/3D tracking accuracy in complex real-world scenes.

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

Table 10: **3D tracking results** on the TAP-Vid3D Benchmark. We use ZoeDepth for depth estimation.[†] denotes using depth to lift 2D tracks to 3D tracks.

| Methods | Aria | | | DriveTrack | | | PStudio | | | Average | | |
|---|---|---|---|---|---|---|---|---|---|---|---|---|
| | AJ↑ | APD$_{3D}$↑ | OA↑ | AJ↑ | APD$_{3D}$↑ | OA↑ | AJ↑ | APD$_{3D}$↑ | OA↑ | AJ↑ | APD$_{3D}$↑ | OA↑ |
| TAPIR[†] + ZoeDepth | 9.0 | 14.3 | 79.7 | 5.2 | 8.8 | 81.6 | 10.7 | 18.2 | 78.7 | 8.3 | 13.8 | 80.0 |
| CoTracker2[†] + ZoeDepth | 10.0 | 15.9 | 87.8 | 5.0 | 9.1 | 82.6 | 11.2 | **19.4** | 80.0 | 8.7 | 14.8 | 83.4 |
| BootsTAPIR[†] + ZoeDepth | 9.9 | 16.3 | 86.5 | 5.4 | 9.2 | 85.3 | **11.3** | 19.0 | 82.7 | 8.8 | 14.8 | 84.8 |
| TAPTR[†] + ZoeDepth | 9.1 | 15.3 | 87.8 | 7.4 | 12.4 | 84.8 | 10.0 | 17.8 | **84.3** | 8.8 | 15.2 | **85.6** |
| LocoTrack[†] + ZoeDepth | 8.9 | 15.1 | 83.5 | 7.5 | 12.3 | 82.8 | 9.7 | 17.1 | 80.1 | 8.7 | 14.8 | 82.1 |
| SpatialTracker + ZoeDepth | 9.2 | 15.1 | 89.9 | 5.8 | 10.2 | 82.0 | 9.8 | 17.7 | 78.0 | 8.3 | 14.3 | 83.3 |
| SceneTracker + ZoeDepth | - | 15.1 | - | - | 5.6 | - | - | 16.3 | - | - | 12.3 | - |
| DELTA + ZoeDepth | **10.1** | **16.2** | 84.7 | 7.8 | 12.8 | 87.2 | 10.2 | 17.8 | 74.5 | **9.4** | 15.6 | 82.1 |
| Ours + ZoeDepth | 10.0 | 15.9 | 86.5 | **8.0** | **13.5** | 87.3 | 9.7 | 18.0 | 73.6 | 9.2 | **15.8** | 81.5 |

Table 11: **Additional 3D tracking results** on the TAP-Vid3D benchmark of recent sparse 2D/3D trackers using depth predicted from Unidepth-V2 (Piccinelli et al., 2024).[†] denotes using depth to lift 2D tracks to 3D tracks.

| Methods | Aria | | | | DriveTrack | | | | PStudio | | | | Average | | | |
|---|---|---|---|---|---|---|---|---|---|---|---|---|---|---|---|---|
| | AJ↑ | APD$_{3D}$↑ | APD$_{3D}^{all}$↑ | OA↑ | AJ↑ | APD$_{3D}$↑ | APD$_{3D}^{all}$↑ | OA↑ | AJ↑ | APD$_{3D}$↑ | APD$_{3D}^{all}$↑ | OA↑ | AJ↑ | APD$_{3D}$↑ | APD$_{3D}^{all}$↑ | OA↑ |
| TrackOn2[†] | **17.2** | **25.5** | 16.9 | **92.4** | 12.5 | 19.2 | 16.7 | **85.4** | **7.8** | 13.9 | 12.2 | **86.7** | 12.5 | 19.5 | 15.2 | **88.2** |
| TAPNext[†] | 15.2 | 22.0 | 13.8 | 81.7 | 14.5 | 22.9 | 20.3 | 76.0 | **7.8** | 14.2 | 12.5 | 83.5 | 12.5 | 19.7 | 15.5 | 80.4 |
| CoTracker3[†] | 15.6 | 23.5 | 15.8 | 90.3 | 14.1 | 21.5 | 18.7 | 80.7 | 7.4 | 13.5 | 11.1 | 80.9 | 12.3 | 19.5 | 15.4 | 83.9 |
| TAPIP3D | 15.9 | 24.3 | 16.8 | 88.6 | 14.1 | 21.4 | 19.9 | 83.0 | 7.4 | 13.5 | 12.1 | 82.4 | 12.4 | 19.7 | 16.2 | 84.6 |
| Ours | 17.0 | 24.7 | **17.6** | 87.2 | **15.6** | **23.8** | **22.5** | 84.6 | 7.7 | **14.4** | **13.2** | 74.1 | **13.4** | **21.0** | **17.8** | 81.2 |

# A  LARGE LANGUAGE MODEL USAGE DISCLOSURE

We used ChatGPT solely for editing the preprint, including grammar checks, phrasing polish, and LaTeX formatting.

# B  EXPERIMENTAL RESULTS

## B.1  EVALUATION METRICS

For the *long-range optical flow*, we adopt the evaluation protocol from Le Moing et al. (2024); Ngo et al. (2024), reporting the end-point error (EPE) between predicted and ground-truth flows and the occlusion prediction accuracy using the intersection-over-union (IoU) between the predicted and ground-truth visibility masks. For the *dense 3D tracking* and *sparse 3D point tracking* benchmarks, we follow the metrics proposed in Koppula et al. (2024). Specifically, we report APD$_{3D}$ (Average Percent of Points within a threshold to measure spatial accuracy), OA (Occlusion Accuracy) for evaluating visibility prediction, and AJ (Average Jaccard), which jointly captures both spatial and occlusion correctness.

## B.2  3D TRACKING

We further evaluate our method on the TAP-Vid3D benchmark using depth maps predicted by ZoeDepth (Bhat et al., 2023). Results are summarized in Tab. 10.

For completeness, we report lifted 3D results for recent sparse 2D trackers (TrackOn2 (Aydemir et al., 2025), TAPNext (Zholus et al., 2025), CoTracker3 (Karaev et al., 2024)) and the new 3D tracker TAPIP3D (Zhang et al., 2025) using UniDepth-v2 in Tab. 11. We also include APD$_{3D}^{all}$—the fraction of points within a 3D error threshold over *both visible and occluded* points—which complements APD$_{3D}$ (visible only). Across TAP-Vid3D, our method consistently surpasses strong 2D trackers on APD$_{3D}$ and APD$_{3D}^{all}$. Our *Occlusion Accuracy* is lower than Aydemir et al. (2025); Zhang et al. (2025), likely because we supervise *both* visible and occluded trajectories, whereas other methods supervise only visible ones (a similar effect appears for CoTracker2 in Tab. 4). Over-

Table 12: **Additional 3D tracking results** on the TAP-Vid3D Benchmark using video depth predicted from Pi3 (Wang et al., 2025). [†] denotes using depth to lift 2D tracks to 3D tracks.

| Methods | Aria | | | | DriveTrack | | | | PStudio | | | | Average | | | |
|---|---|---|---|---|---|---|---|---|---|---|---|---|---|---|---|---|
| | AJ↑ | $APD_{3D}^{vis}$↑ | $APD_{3D}^{all}$↑ | OA↑ | AJ↑ | $APD_{3D}^{vis}$↑ | $APD_{3D}^{all}$↑ | OA↑ | AJ↑ | $APD_{3D}^{vis}$↑ | $APD_{3D}^{all}$↑ | OA↑ | AJ↑ | $APD_{3D}^{vis}$↑ | $APD_{3D}^{all}$↑ | OA↑ |
| TrackOn2[†] | **32.6** | **44.5** | 27.8 | **92.4** | 17.3 | 24.9 | 21.5 | 85.4 | 17.4 | 26.9 | 22.8 | **86.7** | 22.4 | 32.1 | 24.0 | 88.1 |
| TAPNext[†] | 28.9 | 38.8 | 23.4 | 81.7 | 20.0 | 30.3 | 26.5 | 76.0 | 17.3 | 27.4 | 23.3 | 83.5 | 22.1 | 32.1 | 24.4 | 80.4 |
| CoTracker3[†] | 28.6 | 40 | 25.9 | 90.3 | 20.4 | 29.2 | 25.5 | 80.7 | 16.6 | 26 | 22 | 80.9 | 21.9 | 31.7 | 24.5 | 83.9 |
| SpaTracker | 18.9 | 28 | 19.2 | 90.5 | 6.5 | 11.5 | 10.8 | 83.1 | 14.8 | 24.3 | 21.6 | 78.5 | 13.4 | 21.3 | 17.20 | 84.0 |
| TAPIP3D | 30.7 | 42.7 | **29.9** | 90.2 | **22.3** | **30.9** | 29.1 | 84.4 | 18.0 | 27.8 | 24.8 | 83.8 | **23.7** | **33.8** | **27.9** | **86.1** |
| Ours | 30.2 | 40.5 | 27.9 | 88.7 | 21.9 | 30.8 | **29.2** | **85.4** | **18.1** | **28.4** | **25.6** | 80.3 | 23.4 | 33.2 | 27.6 | 84.8 |

Table 13: **2D tracking results** on the TAP-Vid2D Benchmark.

| Methods | DAVIS | | | RGB-Stacking | | |
|---|---|---|---|---|---|---|
| | AJ↑ | $APD_{2D}$↑ | OA↑ | AJ↑ | $APD_{2D}$↑ | OA↑ |
| TAP-Net (Doersch et al., 2022) | 33.0 | 48.6 | 78.8 | 54.6 | 68.3 | 87.7 |
| MFT (Neoral et al., 2024) | 47.3 | 66.8 | 77.8 | - | - | - |
| PIPs (Harley et al., 2022) | 42.2 | 64.8 | 77.7 | 15.7 | 28.4 | 77.1 |
| OmniMotion (Wang et al., 2023a) | 46.4 | 62.7 | 85.3 | 69.5 | 82.5 | 90.3 |
| TAPIR (Doersch et al., 2023) | 56.2 | 70.0 | 86.5 | 54.2 | 69.8 | 84.4 |
| CoTracker2 (Karaev et al., 2023) | 60.6 | 75.4 | 89.3 | 63.1 | 77.0 | 87.8 |
| DOT (Le Moing et al., 2024) | 60.1 | 74.5 | 89.0 | **77.1** | **87.7** | **93.3** |
| BootsTAPIR (Doersch et al., 2024) | 61.4 | 73.6 | 88.7 | 70.8 | 83.0 | 89.9 |
| TAPTR (Li et al., 2024b) | 63.0 | 76.1 | 91.1 | - | - | - |
| TAPTRv2 (Li et al., 2024a) | 63.5 | 75.9 | **91.4** | - | - | - |
| LocoTrack (Cho et al., 2024b) | 63.0 | 75.3 | 87.2 | 69.0 | 83.2 | 89.5 |
| SpatialTracker (Xiao et al., 2024) | 61.1 | 76.3 | 89.5 | 63.5 | 77.6 | 88.2 |
| SceneTracker (Wang et al., 2024a) | - | 71.8 | - | - | 73.3 | - |
| DOT-3D Ngo et al. (2024) | 61.2 | 75.3 | 88.1 | 76.3 | 86.6 | 92.1 |
| DELTA Ngo et al. (2024) | 62.7 | 76.7 | 88.2 | 74.2 | 83.5 | 90.0 |
| Ours | 62.5 | 76.2 | 87.9 | 74.0 | 82.9 | 89.3 |

all, stronger 2D tracker does not guarantee stronger 3D tracking once depth, occlusion, and long-range consistency are considered; moreover, "2D tracking + depth" recovers only visible motion (yielding much lower $APD_{3D}^{all}$), limiting downstream uses (e.g., 3D/4D reconstruction) that require trajectories through occlusions and out-of-frame segments. These findings underscore the need for end-to-end 3D tracking models.

We also report lifted 3D results for recent sparse 2D trackers (TrackOn2 (Aydemir et al., 2025), TAPNext (Zholus et al., 2025), CoTracker3 (Karaev et al., 2024)) and the new 3D tracker TAPIP3D (Zhang et al., 2025), using Pi3 (Wang et al., 2025) video depth (Tab. 12).

## B.3    2D TRACKING

We report the 2D tracking performance on the TAP-Vid2D benchmark (Doersch et al., 2022) in Tab. 13. Our method has comparable performance on the sparse 2D tracking benchmark with recent methods.

## B.4    DENSE 3D TRACKING

Figure 6 shows visual comparisons of dense 3D tracking between our method and baselines. Our model produces highly stable and accurate trajectories, clearly outperforming SceneTracker (Wang et al., 2024a) and SpaTracker (Xiao et al., 2024), and achieving results on par with the strong DELTA (Ngo et al., 2024) baseline.

More qualitative results on dense 3D tracking and comparisons with prior approaches can be found on this webpage https://anonymous40645324.github.io/.

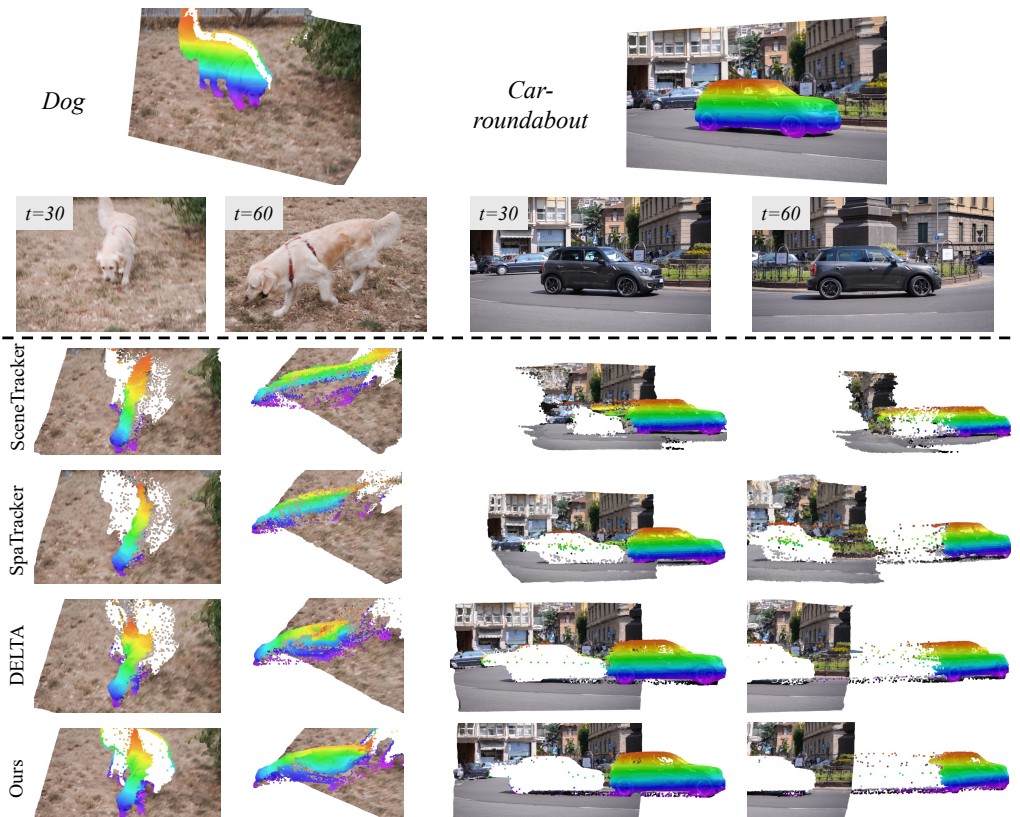

Figure 6: **Qualitative comparison of dense 3D tracking** between SceneTracker (Wang et al., 2024a), SpaTracker (Xiao et al., 2024), DELTA (Ngo et al., 2024), and our method. We track every pixel from the first frame through 3D space, with moving objects highlighted in rainbow colors. Our approach accurately captures foreground motion, preserves stable backgrounds, and operates significantly faster with a $5\times$ to $100\times$ speedup compared to prior methods.

Table 14: Per-module runtime of the final model on an A100 80 GB (16-frame window, $384 \times 512$, 4 iterations, stride schedule $8 \to 4 \to 2 \to 1$). Times in milliseconds.

| Module \ Iter | 1 (s = 8) | 2 (s = 4) | 3 (s = 2) | 4 (s = 1) | Total |
|---|---|---|---|---|---|
| Backbone (once) | 25.0 | — | — | — | 25.0 |
| Transformer | 20.1 | 41.2 | 162.0 | 656.3 | 879.6 |
| 4D Corr (proj-32) | 7.5 | 28.7 | 91.1 | 382.8 | 510.1 |
| Interpolator | 6.1 | 6.1 | 6.1 | — | 18.3 |
| Upsampler (final only) | — | — | — | 362.0 | 362.0 |
| **Frame total** | **58.7** | **76.0** | **259.2** | **1401.1** | **1795.0** |

### B.5 MORE DETAILS ABOUT RUNTIME BREAKDOWN

Table 14 reports a per-module latency breakdown of the final pipeline.

- The early iterations are inexpensive because only $1/64$ and $1/16$ of the pixels are active; most compute is incurred once at $s=1$ for accuracy. This allows tuning the stride schedule (and the number of iterations) to trade runtime for precision.

- The learnable interpolator contributes only a few milliseconds per sparse iteration, yet removes grid artifacts and blurring (see Supp. Figs. 4–7).

Table 15: GPU memory (GB) by iteration and subsampling stride (A100 80 GB, 16-frame window, $384 \times 512$). Our coarse-to-fine uses strides $8 \rightarrow 4 \rightarrow 2 \rightarrow 1$.

| Method | Iter 1 (s=8) | Iter 2 (s=4) | Iter 3 (s=2) | Iter 4 (s=1) |
|---|---|---|---|---|
| DELTA (dense each iter.) | 40.1 | 40.1 | 40.1 | 40.1 |
| Ours (coarse-to-fine) | 25.1 | 27.2 | 32.1 | 40.1 |

- Combined with the projected 4D correlation, the token-level coarse-to-fine schedule yields an overall $\approx 5\times$ speed-up while preserving DELTA-level accuracy.

### B.6 MORE DETAILS ABOUT MEMORY CONSUMPTION

We profiled GPU memory across iterations under our coarse-to-fine schedule ($8 \rightarrow 4 \rightarrow 2 \rightarrow 1$) and compared it with DELTA (Ngo et al., 2024), which tracks all low-resolution pixels at every refinement step (Table 15). Because we densify to $s=1$ in the last iteration to reach pixel-level accuracy, the *peak* memory equals DELTA's (40.1 GB). However, earlier iterations are **37.4%**, **32.2%**, and **20.0%** lower than DELTA (25.1/27.2/32.1 vs. 40.1 GB at iters 1–3), complementing the $\sim 5\times$ runtime gain reported in the paper. For sparse trackers such as SpatialTracker (Xiao et al., 2024), CoTracker2 (Karaev et al., 2023), and LocoTrack (Cho et al., 2024b), a single dense pass over all pixels and long sequences does not fit on an 80GB GPU in our evaluation setting.

## C ANALYSIS OF COARSE-TO-FINE DENSE TRACKING

### C.1 FURTHER ANALYSIS OF STRATEGIES FOR ACCELERATING TRACKING

Fig. 7 illustrates the conceptual overview of each acceleration strategy described in Sec. 3.2 of the main paper. To better understand their practical impact, we further evaluate these strategies across a broader range of hyperparameters:

**1) Spatial downsampling of RGB input:** applied with spatial reduction factors of $2\times$, $4\times$, and $8\times$.

**2) Temporal subsampling of input frames:** applied with temporal reduction factors of $2\times$, $4\times$, and $8\times$.

**3) Trajectory point subsampling:** applied with trajectory subsampling ratios of $4\times$, $16\times$, and $64\times$.

Each variant is followed by upsampling to the original spatial and temporal resolution using either (1) bilinear interpolation or (2) nearest-neighbor interpolation. The corresponding quantitative results are summarized in Tab. 16.

### C.2 SPECTRAL ANALYSIS OF THE OPTICAL FLOW PREDICTIONS

To analyze the spatial frequency characteristics of optical flow, we follow prior work (Skorokhodov et al., 2025) on spectral analysis using the two-dimensional discrete cosine transform (2D DCT) (Ahmed et al., 2006). Given a flow field $\mathcal{F}_{flow} \in \mathbb{R}^{H \times W \times 2}$, we process the horizontal and vertical components separately by dividing each into non-overlapping blocks of size $B \times B$. We then apply the type-II 2D DCT to each block, which projects the flow values onto a set of cosine basis functions at different spatial frequencies. Within each block, we arrange the coefficients into a one-dimensional sequence using the standard JPEG zigzag ordering (Wallace, 1991), which traverses from low to high spatial frequencies. To summarize the overall spectral content of the flow, we average these zigzag-ordered coefficients across all blocks and across both flow components (horizontal and vertical). This produces a compact frequency profile that captures the distribution of motion energy across spatial scales.

We apply this process independently to the predicted optical flow at each iteration of the model, as well as to the ground-truth flow. This allows us to track how the spectral properties of the predicted flow evolve during refinement and how they compare to the underlying ground-truth signal.

Table 16: Comparison of cost reduction strategies on the Kubric3D *val* set (Ngo et al., 2024). All methods use 1 iteration unless noted; full-resolution outputs are obtained using either bilinear or nearest-neighbor interpolation.

| Baseline | APD$_{3D}$ $\uparrow$ | | | | Runtime$\downarrow$ | |
|---|---|---|---|---|---|---|
| DELTA (4 iterations) | 87.3 | | | | 8404 | |
| DELTA (1 iteration) | 74.3 | | | | 2275 | |
| **Strategy** | 2$\times$ | | 4$\times$ | | 8$\times$ | |
| | APD$_{3D}$ $\uparrow$ | Runtime$\downarrow$ | APD$_{3D}$ $\uparrow$ | Runtime$\downarrow$ | APD$_{3D}$ $\uparrow$ | Runtime$\downarrow$ |
| **(a) Bilinear interpolation** | | | | | | |
| Downsample video reso. | 54.7 | 759 | 35.8 | 383 | 17.8 | 214 |
| Subsample video frames | 70.8 | 1119 | 65.4 | 833 | 42.3 | 491 |
| Subsample trajectories | 72.9 | 917 | 71.2 | 585 | 68.5 | 406 |
| **(b) Nearest-neighbor interpolation** | | | | | | |
| Downsample video reso. | 53.9 | 748 | 34.4 | 382 | 17.6 | 212 |
| Subsample video frames | 68.8 | 1187 | 60.0 | 821 | 49.3 | 487 |
| Subsample trajectories | 73.7 | 902 | 72.1 | 580 | 67.6 | 405 |

The DCT spectrum is shown in Fig. 8, revealing a trade-off introduced by the coarse-to-fine strategy. While this approach helps accelerate inference and preserve low-frequency motion structures, it tends to suppress high-frequency components, which are essential for capturing fine-grained motion details and sharp flow boundaries. In contrast, the model without coarse-to-fine better preserves high-frequency content and more closely matches the spectral distribution of the ground-truth flow across all frequency bands. Notably, this limitation of the coarse-to-fine model is largely mitigated in the final iteration, where full-resolution trajectories are used. As shown in the left plot, the spectrum of the last iteration closely aligns with the ground truth, indicating that fine details can still be recovered at the final refinement stage.

We visualize the predicted optical flow and corresponding error maps across different transformer iterations in Fig. 9,10,11,12, comparing models with and without the coarse-to-fine strategy and using different interpolation methods. The results show that nearest-neighbor interpolation introduces noticeable grid artifacts, while bilinear interpolation tends to oversmooth motion boundaries. In contrast, our proposed learnable interpolator yields more accurate and spatially coherent predictions, particularly in early iterations where only a sparse grid of points is tracked and interpolation plays a critical role in producing dense motion estimates.

## D ANALYSIS ON THE EFFECTIVENESS OF LEARNABLE INTERPOLATOR

We assess interpolation quality at two different stages in the pipeline (using the CVO test set and 8→4→2→1 schedule as in Sec. 4.2): pre-upsampler (low-resolution), i.e., immediately after iterative refinement at stride $s > 1$; and post-upsampler (full-resolution), i.e., after DELTA's attention-based upsampler (as in Fig. 5b). Results are reported in Table 17. At low resolution (Table 17a), performance reflects the interpolator directly; after upsampling (Table 17b), results reflect how well the interpolated seeds support the attention upsampler.

In pre-upsampler, NN interpolator preserves sharp edges but is blocky , while bilinear one is smoother but blurs boundaries; their relative ranking can fluctuate with stride and scene content. In post-upsampler, DELTA's attention module suppresses nearest's local artifacts but cannot restore high-frequency detail erased by bilinear's averaging, so nearest interpolator typically outperforms the bilinear mode. In both regimes, our feature-aware interpolator preserves boundaries and avoids NN blockiness by adapting weights to appearance similarity, so it's robust pre-upsampler and provides better post-upsampler results, yielding the most reliable accuracy.

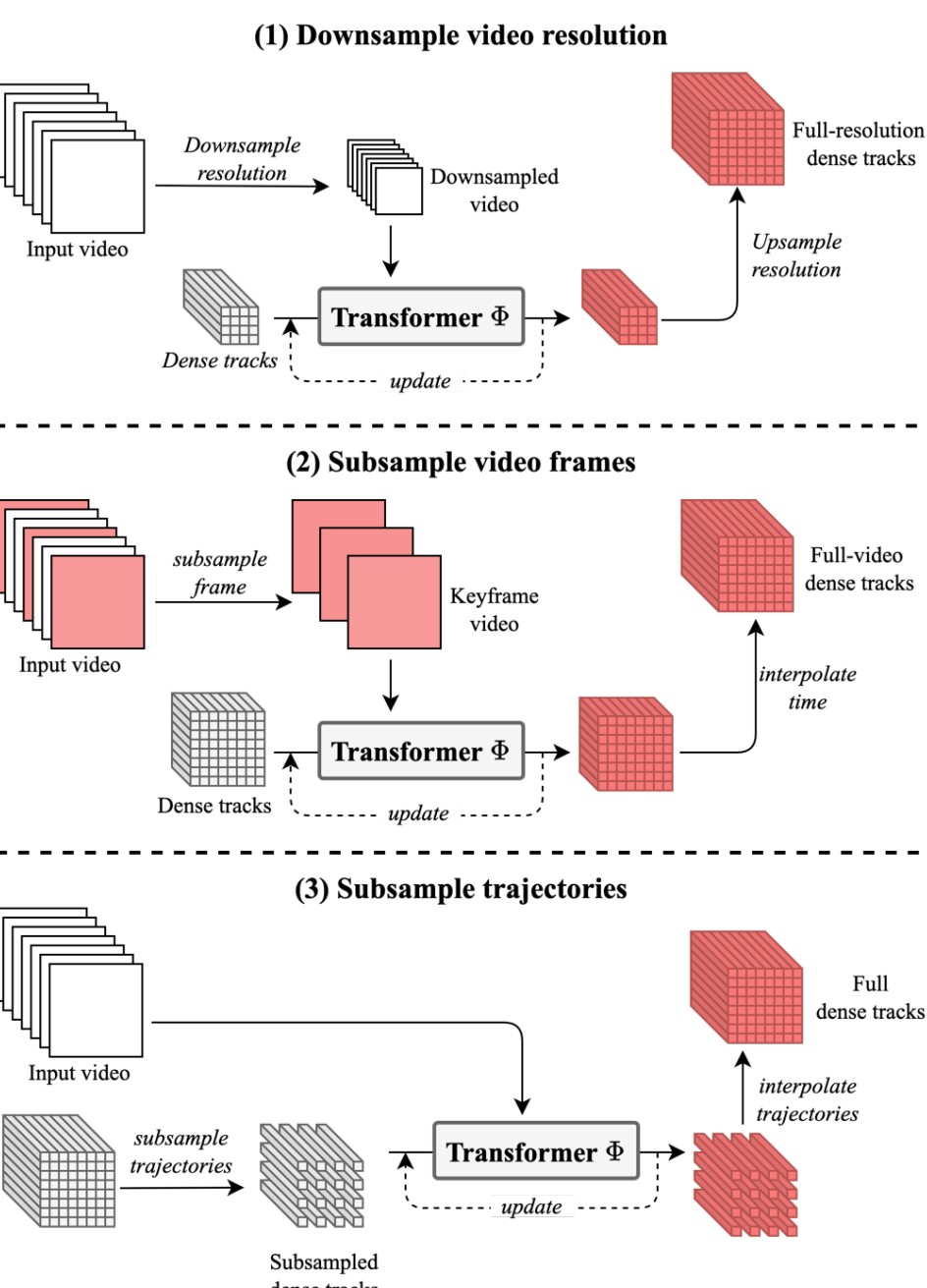

Figure 7: Illustration of three acceleration strategies. (1): Spatial downsampling of RGB input; (2) Temporal subsampling of input ; (3) Trajectory point subsampling;

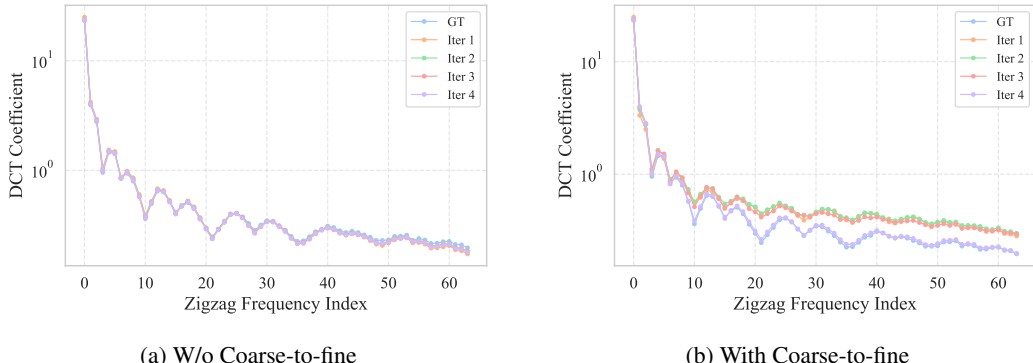

(a) W/o Coarse-to-fine

(b) With Coarse-to-fine

Figure 8: DCT spectrum of predicted optical flows across iterations and ground truth: (a) without coarse-to-fine, (b) with coarse-to-fine.

Table 17: End-point-error (EPE↓) under coarse-to-fine strategy before versus after the upsampler.

(a) Pre-upsampler (low-resolution)

| Iter | Learnable | Nearest | Bilinear |
|------|-----------|---------|----------|
| 1    | 1.34      | 2.39    | 2.35     |
| 2    | 0.85      | 1.32    | 1.36     |
| 3    | 0.66      | 0.77    | 0.82     |
| 4    | 0.46      | 0.50    | 0.47     |

(b) Post-upsampler (full-resolution)

| Iter | Learnable | Nearest | Bilinear |
|------|-----------|---------|----------|
| 1    | 10.62     | 11.10   | 12.54    |
| 2    | 5.61      | 5.89    | 6.21     |
| 3    | 4.06      | 4.23    | 4.72     |
| 4    | **3.57**  | 3.66    | 3.95     |

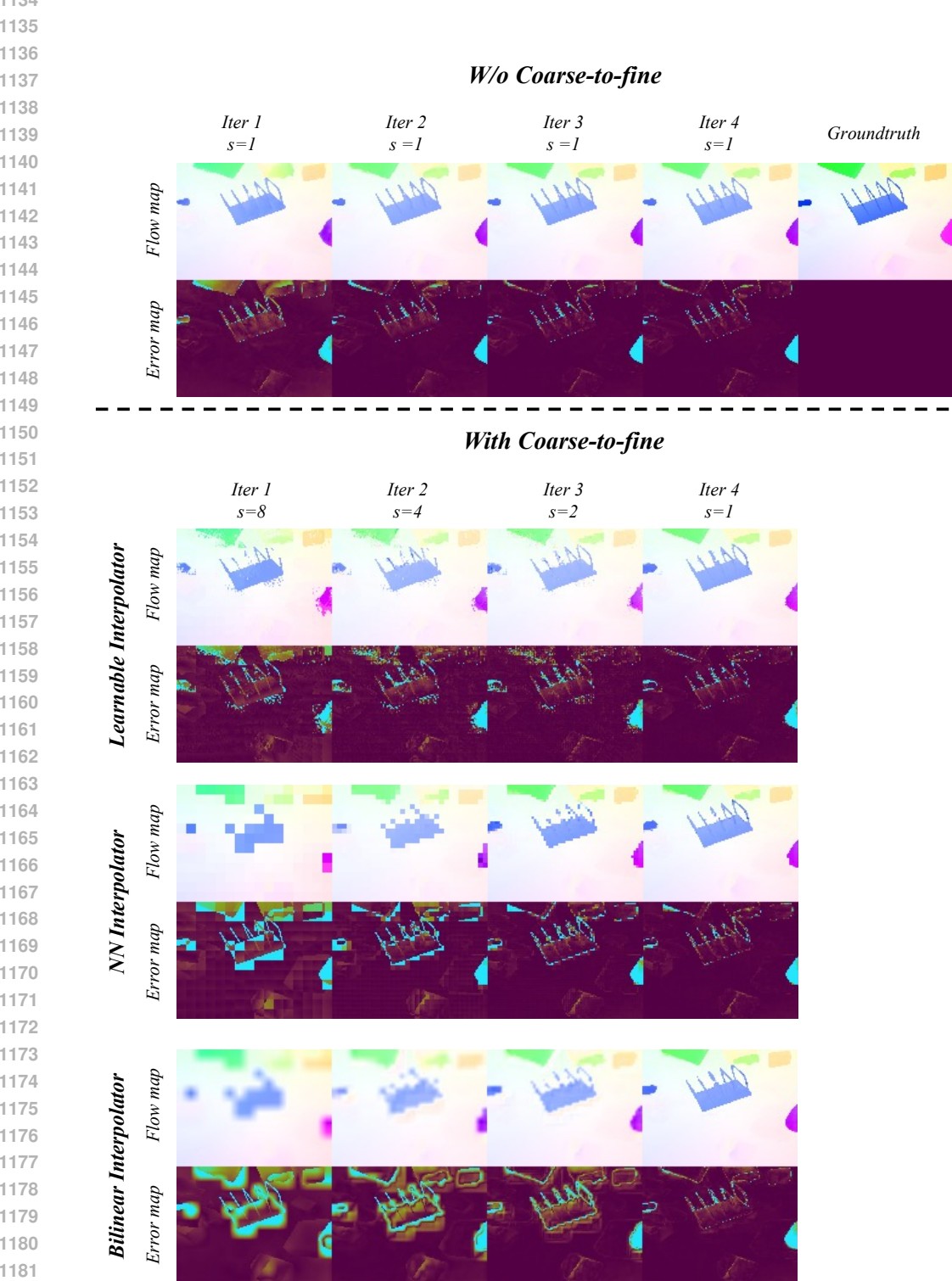

Figure 9: Visualization of predicted optical flow and corresponding error maps across different iterations. We compare models with and without the coarse-to-fine strategy, using various interpolation methods.

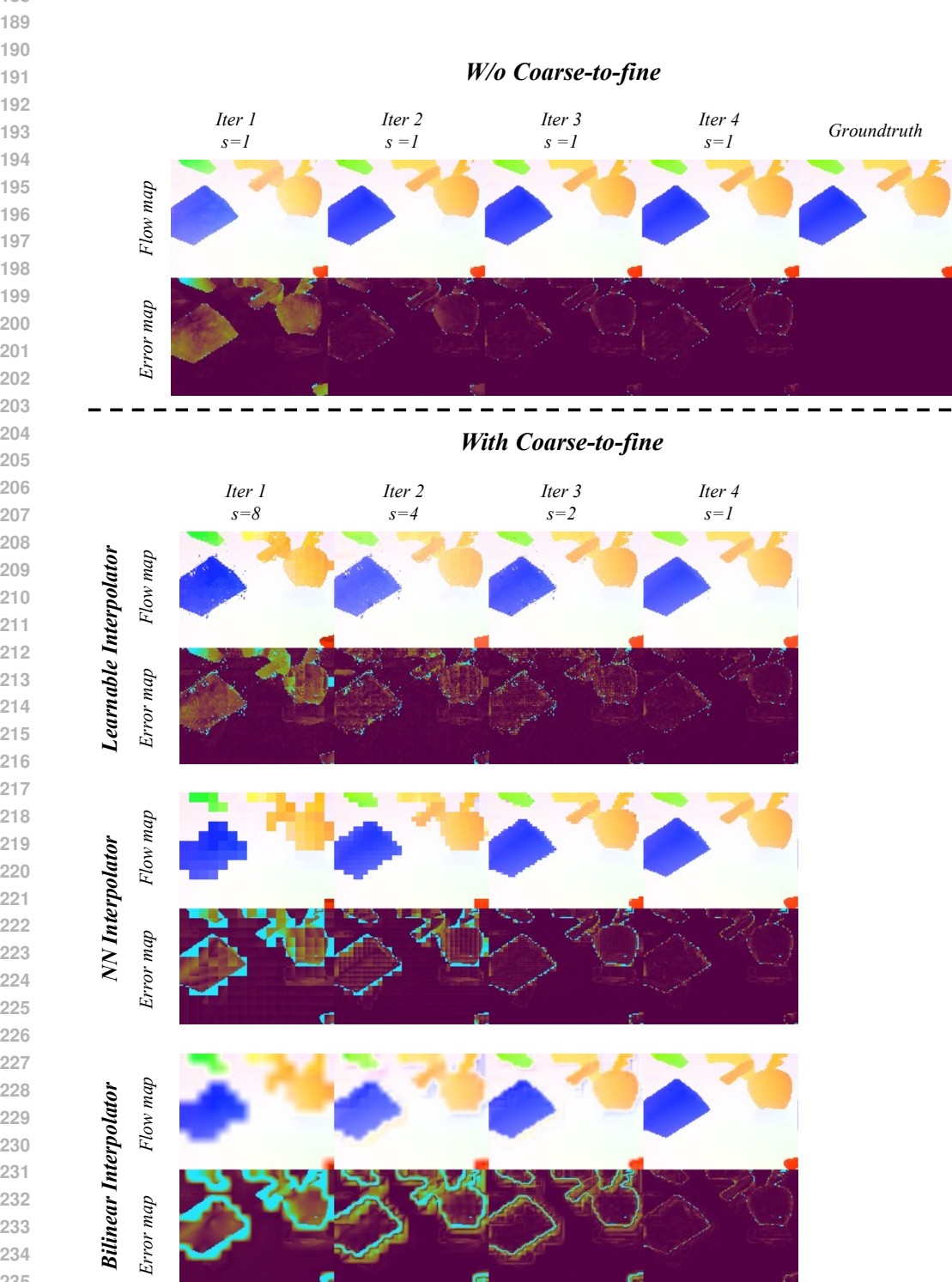

Figure 10: Visualization of predicted optical flow and corresponding error maps across different iterations. We compare models with and without the coarse-to-fine strategy, using various interpolation methods.

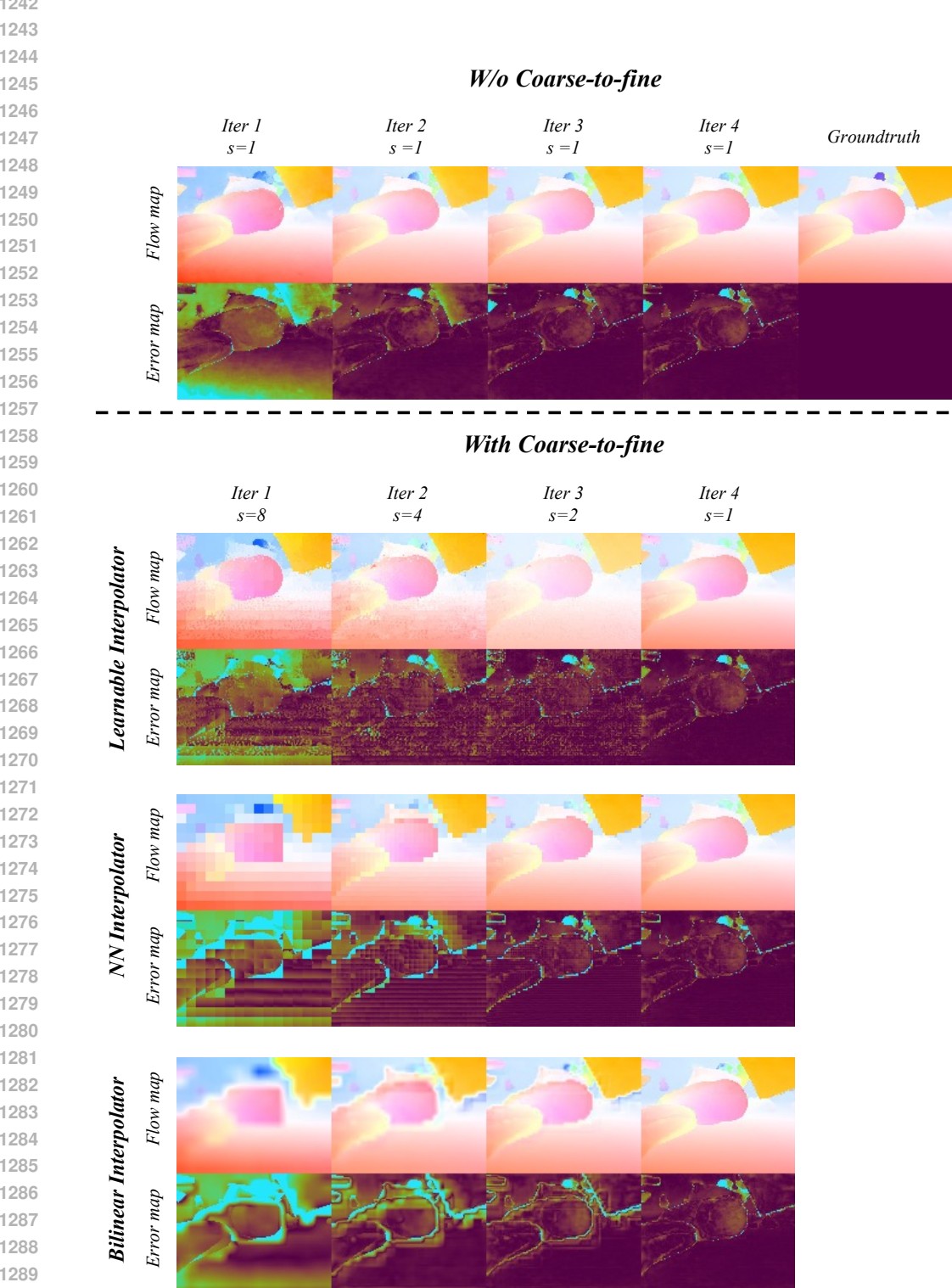

Figure 11: Visualization of predicted optical flow and corresponding error maps across different iterations. We compare models with and without the coarse-to-fine strategy, using various interpolation methods.

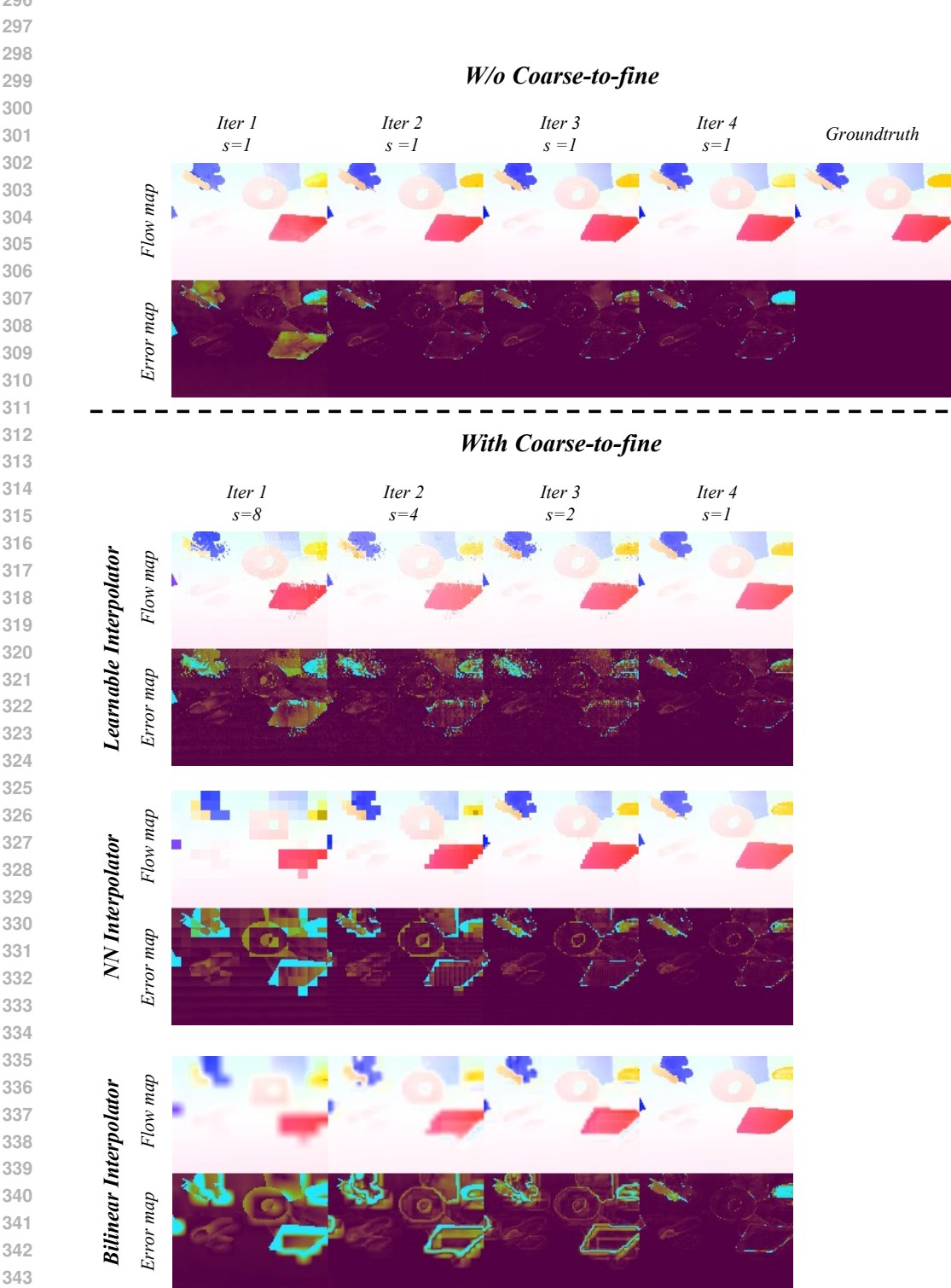

Figure 12: Visualization of predicted optical flow and corresponding error maps across different iterations. We compare models with and without the coarse-to-fine strategy, using various interpolation methods.

