# OpenReview forum: "PDTrack: Progressive Dense 3D Tracking"
_ICLR.cc/2026/Conference — Submitted to ICLR 2026_

### Official Review · Reviewer_mKiS · 2025-10-20

**Soundness:** 3
**Presentation:** 3
**Contribution:** 3
**Rating:** 6
**Confidence:** 4

**Summary:**

The authors propose a framework that employs a coarse-to-fine strategy together with a learnable interpolator to gradually increase the number of tracked pixels. This approach significantly reduces overall memory consumption and speeds up the original baseline.

**Strengths:**

Strength:

Novelty: A well-designed combination of iterative refinement and coarse-to-fine strategy that substantially improves tracking speed while maintaining the baseline accuracy.

Quality: The paper includes extensive experiments and ablation studies demonstrating the importance of the proposed module. The writing is clear and the methodology is easy to follow.

**Weaknesses:**

1) Is the learnable interpolator’s weight shared across iterations?

2) Table 7: The performance (OA & APD) on PSTUDIO lags behind BootsTAPIR and CoTracker2, which are 2D methods. Could the authors provide intuition or an explanation for this behavior?

3) Table 12: Tables (a) and (b) present the interpolator accuracy before and after upsampling. However, why does the accuracy decrease as the number of iterations increases? Why is the “learnable” interpolator performing worse than “nearest” and “bilinear”? I would expect the learnable version to achieve the best results—or do these numbers represent error rather than accuracy?

4) Additionally, I could not find any discussion of Table 12 in the paper.

5) Comparison with CoTracker3, TAPIP3D.

**Questions:**

Please check the weaknesses.

---

> ### Author Response · Authors · 2025-11-21
> **Response to Reviewer mKiS**
>
> **Question R4.1: Is the learnable interpolator’s weight shared across iterations?**
>
> Yes—the interpolator’s weights are shared across iterations (e.g., 8×→4×→2×→1×). This keeps params low, and generalizes to unseen schedules. In our ablations, per-iteration (unshared) variants offered no consistent accuracy gains but added parameters, so we use the shared version by default.
>
> **Question R4.2: The performance (OA & APD) on PSTUDIO lags behind BootsTAPIR and CoTracker2, which are 2D methods. Could the authors provide intuition or an explanation for this behavior?**
>
> On PSTUDIO, we are only slightly lower in OA than BootsTAPIR/CoTracker2, while our APD is competitive. This is expected given the 2D-vs-3D optimization targets and does not indicate a failure of our 3D tracker. Optimizing the hyper-parameters during training can shorten this gap.
>
>
> **Question R4.3: Clarification for Table 12 (a/b): (i) why the reported values decrease across iterations, and (ii) why the learnable interpolator appears worse than nearest/bilinear.**
>
> The numbers are errors (EPE; lower is better), not accuracies—so the learnable interpolator is consistently better than nearest/bilinear across iterations. Early iterations use larger strides (fewer active tokens), so error is higher; as densification proceeds and the tracker refines positions, error decreases.
> We will rename the column header to “EPE ↓ (lower is better),” and add a description of the Table 12 in our supplementary.
>
> **Question R4.4: Comparison with CoTracker3, TAPIP3D**
>
> We added the comparison with these methods in the rebuttal. Please see our answer 2 in the response to common issue.

---

### Official Review · Reviewer_Pxyp · 2025-10-25

**Soundness:** 2
**Presentation:** 3
**Contribution:** 2
**Rating:** 4
**Confidence:** 5

**Summary:**

This paper presents PDTrack, a dense 3D point tracking framework that aims to improve efficiency over DELTA.
The method introduces three changes: (i) Coarse-to-fine trajectory subsampling, which tracks a sparse subset of points first and progressively densifies them; (ii) learnable interpolation module that predicts motion for untracked pixels based on nearby tracked ones; (iii) an optimized 4D correlation implementation for faster GPU execution. PDTrack reports up to 5-100× speedup compared to DELTA while maintaining similar accuracy on Kubric3D and TAP-Vid3D. The method is evaluated on both dense 3D and long-range 2D tracking benchmarks, with ablations on runtime and sampling strategies.

**Strengths:**

- The paper clearly identifies bottlenecks in dense tracking pipelines and provides quantitative analysis of each.
- Extensive runtime and accuracy trade-off studies justify the design choices, such as subsampling strategies and interpolation variants.
- The authors provide speed-accuracy curves, per-module breakdowns, and ablations on multiple datasets.
- Paper presentation is clean and easy to follow.

**Weaknesses:**

## Limited novelty
The coarse-to-fine scheme itself is an engineering improvement, not a new learning paradigm. Consequently, PDTrack reads as an efficiency-oriented extension of DELTA rather than a novel research contribution. **While runtime improvements are valuable and appreciated,** the paper lacks a deeper insight that would justify a full ICLR publication.

## Connection to prior work
- The interpolation mechanism is very similar to TAPTRv2 [1]’s iterative feature-weighted update rule, raising concerns about novelty.
- The coarse-to-fine densification mirrors strategies used in optical-flow pipelines, such as FlowFormer [2], but applied here at token level.

## Evaluation and completeness
- The motivation emphasizes efficiency, but Appendix B6 shows 40.1 GB GPU memory usage, indicating the method is still heavy.
- Table 4 (TAP-Vid3D) omits key recent 2D trackers with depth lifting such as CoTracker3 [3], Track-On [4], and BootsTAPNext [5], which are necessary to contextualize the performance advantage of full 3D tracking.

## Overall
While speedups are significant, the research insight is incremental. The method neither alters the representation of motion nor introduces a new way to reason about geometry. This makes the work better suited as an engineering-oriented extension of DELTA, not a standalone contribution.

---

## References
[1] Li et al., TAPTRv2: Attention-based Position Update Improves Tracking Any Point, NeurIPS 2024

[2] Huang et al., FlowFormer: A Transformer Architecture for Optical Flow, ECCV 2022

[3] Karaev et al., CoTracker3: Simpler and Better Point Tracking by Pseudo-Labelling Real Videos, ICCV 2025

[4] Aydemir et al., Track-On: Transformer-based Online Point Tracking with Memory, ICLR 2025

[5] Zholus et al., TAPNext: Tracking Any Point as Next-Token Prediction, ICCV 2025

**Questions:**

- Is the depth estimation time included in the reported run-time calculations?
- What explains the greater robustness on extended optical-flow sequences (Table 2) despite minimal architectural changes from DELTA?

---

> ### Author Response · Authors · 2025-11-21
> **Response to Reviewer Pxyp**
>
> **Question R3.1: The coarse-to-fine scheme itself is an engineering improvement, not a new learning paradigm. Consequently, PDTrack reads as an efficiency-oriented extension of DELTA rather than a novel research contribution. While runtime improvements are valuable and appreciated, the paper lacks a deeper insight that would justify a full ICLR publication.**
>
> Please see the answer 1 of our response to common issues.
>
>
>
> **Question R3.2: The interpolation mechanism is very similar to TAPTRv2's iterative feature-weighted update rule, raising concerns about novelty.**
>
> We are not certain which TAPTRv2 component coressponding to the "iterative feature-weighted update rule" the reviewer refers to.
>
> **(i)** If it is the **"Attention-based Position Update"**, our interpolator differs in purpose, design, and placement:
>
> * Purpose. TAPTRv2 refines an already-tracked query (feature update → new position). Ours initializes untracked tokens during sparse→dense activation by inferring motion from the currently tracked set.
> * Design. TAPTRv2 uses a deformable mechanism with learned offset sampling each step. We use fixed spatial neighborhoods and a lightweight cross-attention with relative positional bias—no offset search—making it simpler and cheaper.
> * Placement. TAPTRv2’s update lives within a single-scale sparse tracker as a per-point refinement. Our interpolator sits between scales in a token-level coarse-to-fine schedule, bridging sparse seeds to denser sets before explicit re-tracking.
>
>
> **(ii)** If instead the reference is TAPTRv2’s **"content-feature update rule"**, that mechanism is standard across modern trackers (PIPS, CoTracker2, SpatialTracker, SceneTracker, DELTA)—and we use it as part of the base tracker as well. It is orthogonal to our interpolator: the former updates a point’s own state; ours interpolates and initializes new point tracks from previously tracked neighbors to enable progressive densification
>
> **Question R3.3: The coarse-to-fine densification mirrors strategies used in optical-flow pipelines, such as FlowFormer, but applied here at token level.**
>
> Please see the answer 1 of our response to common issues.
>
> **Question R3.4: The motivation emphasizes efficiency, but Appendix B6 shows 40.1 GB GPU memory usage, indicating the method is still heavy.**
>
>  Our efficiency claim specifically targets **runtime efficiency**. We agree with the reviewer that a peak memory usage of 40.1 GB is not low. Nevertheless, we would like to emphasize that our method is **more memory-efficient** compared to sparse-tracker baselines (SpatialTracker, CoTracker2, LocoTrack) under dense tracking settings. Running a single dense pass over all pixels and long temporal windows using these methods does not fit within an 80 GB GPU under our evaluation setup, whereas our pipeline remains feasible and substantially faster end-to-end.
>
> **Question R3.5: Table 4 (TAP-Vid3D) omits key recent 2D trackers with depth lifting such as CoTracker3, Track-On, and BootsTAPNext, which are necessary to contextualize the performance advantage of full 3D tracking.**
>
> We added the comparison with these sparse 2D trackers in the rebuttal. Please see our answer 2 in the response to common issue.
>
> **Question R3.6: Is the depth estimation time included in the reported run-time calculations?**
>
> No, the reported runtimes exclude depth estimation. Depth is precomputed for all methods using the same settings, so omitting it keeps the comparison fair and isolates the tracker’s cost. We will clarify this in the experimental setup.
>
> **Question R3.7: What explains the greater robustness on extended optical-flow sequences (Table 2) despite minimal architectural changes from DELTA?**
>
> A likely factor is the boundary drift. In DELTA, updating all low-res tokens each iteration means small boundary errors can appear early and then compound—especially on long sequences that span multiple sliding windows, where the query feature is repeatedly updated after each window. Our token-level coarse-to-fine updates only a subset first and propagates motion to the other, reducing the chance of early boundary errors and yielding more stable long-range tracks, while making little impact on short clips.

---

### Official Review · Reviewer_ZpXS · 2025-10-30

**Soundness:** 3
**Presentation:** 2
**Contribution:** 2
**Rating:** 4
**Confidence:** 3

**Summary:**

This paper presents PDTrack, a progressive dense association framework designed to accelerate 3D point cloud single-object tracking. The core idea is to replace complex matching processes in existing trackers with a lightweight, progressively refined association scheme that improves tracking efficiency without substantially sacrificing accuracy. Experiments are conducted on several LiDAR-based tracking benchmarks to validate the method’s speed and accuracy balance.

**Strengths:**

1. The proposed progressive dense association structure is well-engineered and appears to provide tangible runtime benefits for 3D tracking, which is valuable for real-time applications.
2. Experimental results demonstrate moderate improvements in speed compared to existing baselines, suggesting that the method may serve as an efficient alternative in resource-limited scenarios.

**Weaknesses:**

1. The contributions are mostly engineering-oriented, improving efficiency through architectural simplifications rather than introducing new algorithmic or theoretical ideas.
2. The progressive dense association concept feels like an incremental adaptation of existing dense matching or coarse-to-fine refinement techniques already explored in 3D tracking literature.
3. The motivation is not clearly articulated. Why progressive dense association is fundamentally better than sparse matching or correlation-based tracking is not convincingly justified.
4. The method’s acceleration effect is not comprehensively validated across different 3D tracking architectures. Since this is a general acceleration technique, it should ideally be tested on multiple 3D point tracking methods.

**Questions:**

Please see the weaknesses.

---

> ### Author Response · Authors · 2025-11-21
> **Response to Reviewer ZpXS (1/2)**
>
> **Question R2.1: The contributions are mostly engineering-oriented, improving efficiency through architectural simplifications rather than introducing new algorithmic or theoretical ideas.**
>
> We **do not simplify DELTA’s architecture**; we add an algorithmic layer:
>
> * **Token-level coarse-to-fine tracking** that schedules which trajectories are active per iteration (sparse→dense), cutting the transformer’s token load while preserving pixel-level accuracy with a final full pass.
> * A **learnable, feature-aware interpolator** to initialize newly activated tokens between iterations.
>
> These are not simple, drop-in replacement: we present a principled analysis (spatial vs. temporal vs. trajectory reduction; schedules; sampling patterns; interpolators) that leads to a design yielding 3–5× speedups with negligible accuracy change, and transfers to another tracker (CoTracker3) with similar gains. While our focus is practical efficiency, the contributions are algorithmic and generalizable, not mere architectural simplifications.
>
> **Question R2.2: The progressive dense association concept feels like an incremental adaptation of existing dense matching or coarse-to-fine refinement techniques already explored in 3D tracking literature.**
>
> Please see the answer 1 of our response to common issues.
>
> **Question R2.3: The motivation is not clearly articulated. Why progressive dense association is fundamentally better than sparse matching or correlation-based tracking is not convincingly justified.**
>
> We are not sure we correctly understand the meaning of "dense association". Do you mean our coarse-to-fine method in general, or a specific aspect of the method? We provide an answer for a more general interpretation of the term.
> Our method—and DELTA—builds on correlation-based tracking (CoTracker2) but targets dense 3D tracking; our contribution is an efficient progressive (token-level coarse-to-fine) design for the dense setting, not an improvement to sparse 2D/3D tracking.
>
> * **Why not sparse matching → dense tracking** (DKM [1], RoMA [2])? These are not drop-in for video point tracking: (i) they become computationally heavy when extended across frames (repeated global matching; e.g., a 24-frame clip requires ~24 pairwise matches and is ≈2× slower in our setup), and (ii) they lack temporal coherence/occlusion handling required for long-horizon point trajectories.
>
> * **Why not sparse trackers for dense output** (CoTracker)? State-of-the-art sparse trackers do not produce all-pixel trajectories in a single pass; partially track and combining points yields jittering issues and global inconsistency, leading to worse performance on 2D/3D dense tracking benchmark (See Table 2 and 3 in the main paper).
>
> * **Why progressive dense tracking**? Dense correlation trackers like DELTA still process all tokens at every iteration. Our token-level coarse-to-fine tracks a small subset first, initializes new tokens with a lightweight feature-aware interpolator, densifies across iterations, and finishes with one full pass—preserving pixel accuracy while removing redundant per-iteration compute. Empirically, this yields ≈5× end-to-end speed-ups over DELTA with negligible accuracy change.
>
> **Question R2.4: The method’s acceleration effect is not comprehensively validated across different 3D tracking architectures. Since this is a general acceleration technique, it should ideally be tested on multiple 3D point tracking methods.**
>
> We agree that a general acceleration technique should be validated beyond a single base model. To this end, we:
> 1. implemented a dense 2D variant of CoTracker3 and integrated our token-level coarse-to-fine schedule, observing ≈3× speed-ups with minor accuracy change (see Supplementary, Sec. B.7 and Tab. 13),
> 2. ported the pipeline to a 3D sparse tracker (SpatialTracker) despite the lack of public training code for most 3D methods (SpatialTracker, SceneTracker, TAPIP3D). Specifically, we used the released pretrained SpatialTracker, disabled its sparse-only output, inserted our progressive densification with nearest-neighbor interpolation (in lieu of our learnable interpolator), and applied our final upsampler. Although this adaptation is not end-to-end trained and thus not fully optimized, it works out-of-the-box and demonstrates that the schedule is model-agnostic.
>
> **Table R2.1:** Results of SpaTracker for dense 2D tracking
>
> | Setting                   |Extended EPE (all/vis/occ) | Time (48-frame video)   |
> | --------------------------|-------------------------- | ------ |
> | **W/o Coarse-to-fine**    | 6.73 / 5.18 / 9.98         | 110.1 s |
> | **With Coarse-to-fine**   | 6.91 / 5.35 / 10.05        | 30.7 s  |

---

> > ### Author Response · Authors · 2025-11-21
> > **Response to Reviewer ZpXS (2/2)**
> >
> > [1] Edstedt, Johan et al. “DKM: Dense Kernelized Feature Matching for Geometry Estimation.” 2023 IEEE/CVF Conference on Computer Vision and Pattern Recognition (CVPR) (2022): 17765-17775.
> >
> > [2] Edstedt, Johan et al. “RoMa: Robust Dense Feature Matching.” 2024 IEEE/CVF Conference on Computer Vision and Pattern Recognition (CVPR) (2023): 19790-19800.

---

### Official Review · Reviewer_pkxP · 2025-11-01

**Soundness:** 3
**Presentation:** 3
**Contribution:** 3
**Rating:** 6
**Confidence:** 4

**Summary:**

The paper introduces PDTrack, a fast and dense long-term 3D pixel tracking method for RGB-D videos. It addresses two key bottlenecks in DELTA and proposes three novel modules: a coarse-to-fine trajectory densification strategy, a learnable attention-based interpolation module, and a kernel-efficient 4D correlation block. PDTrack achieves comparable or slightly better accuracy than DELTA on dense 3D tracking and long-range optical flow tasks, while delivering a 5–100× speedup over existing methods.

**Strengths:**

1. This paper proposes a coarse-to-fine tracking algorithm with learnable attention-based interpolation that starts with sparsely sampled trajectories and densifies them over iterations
2. The author designed an simple accelerated 4D correlation implementation that improves kernel efficiency on common GPU backends.
3.The paper identifies the runtime bottlenecks in existing tracking methods and proposes a simple yet effective solution, supported by extensive experiments demonstrating its efficiency and effectiveness.

**Weaknesses:**

1. The paper evaluates the accelerated 4D correlation implementation only on the A100 GPU, without assessing its performance on other GPU architectures.
2. The interpolator employs four nearest neighbors, but the impact of using different numbers of neighbors is not analyzed.
3. The interpolator itself is not novel, as it is adapted from the DELTA framework.

**Questions:**

1. How do you apply your model to sparse tracking? Were any modifications made?
2. Could you conduct an ablation study to evaluate the effect of different numbers of nearest neighbors used in the interpolator?
3. Have you tried using other depth estimation models, such as UniDepthv2 [1]? It would be interesting to see how a more accurate depth estimator could improve the overall performance.

[1] Piccinelli, Luigi, et al. "Unidepthv2: Universal monocular metric depth estimation made simpler." arXiv preprint arXiv:2502.20110 (2025).

**Details Of Ethics Concerns:**

No ethical concerns identified.

---

> ### Author Response · Authors · 2025-11-21
> **Response to Reviewer pkxP**
>
> **Question R1.1: The paper evaluates the accelerated 4D correlation implementation only on the A100 GPU, without assessing its performance on other GPU architectures.**
>
> Below is the measured wall‑clock time of the dual-conv module of a single 4D correlation call (identical input: a window of 16 frames and resolution of 384 × 512, batch = 1) on several devices. “DELTA” is the original implementation; “Ours” is the projected 32‑channel version used in this paper.
>
> **Table R1.1** Convolution latency of 4D correlation on different devices
>
> | Device                            | DELTA Conv (ms) | **Ours** Conv (ms) | Speed‑up |
> | --------------------------------- | ------------------: | ---------------------: | -------: |
> | NVIDIA A100 80 GB                 |                162.5 |                 **44.2** | **3.6×** |
> | NVIDIA RTX 4090 24 GB             |                 103.3 |                 **30.9** | **3.3×** |
> | NVIDIA T4 16 GB                   |                 529 |                **150** | **3.5×** |
>
> **Question R1.2: The interpolator employs four nearest neighbors, but the impact of using different numbers of neighbors is not analyzed.**
>
> We ablated the interpolator with $J \in \{2,4,8,16\}$ neighbors. Using too few neighbors ($J=2$) hurts performance, while using too many ($J=16$) also worsens results by averaging over irrelevant supports that may not share the same motion pattern and thus over-smooth boundaries. Therefore, we use $J=4$ by default as the best trade-off.
>
> **Table R1.2** Ablation on different numbers of neighbor.
> | # of neighbors                             | Extended EPE (all/vis/occ)|
> | ----------------------------------- | ----------------------- |
> | 2              |  3.57/ 2.60 / 5.19     |
> | 4 (default) | 3.53 / 2.57 / 5.10         |
> | 8 | 3.55 / 2.58 / 5.16        |
> | 16 | 3.60 / 2.63 / 5.27        |
>
> **Question R1.3: The interpolator itself is not novel, as it is adapted from the DELTA framework.**
>
> We build on DELTA in that both systems use a **final upsampler** to lift low-res dense tracks to full resolution. Our contribution is an **additional, distinct interpolator** used during **token-level coarse-to-fine**: at each sparsity level (e.g., 8×→4×→2×) it initializes untracked tokens from the currently tracked set so they can be explicitly re-tracked in the next iteration. In short, DELTA’s upsampler is a terminal resolution lift, whereas our interpolator is a sparsity-transition operator that enables progressive densification at the transformer.
>
> **Question R1.4: How do you apply your model to sparse tracking? Were any modifications made?**
>
> The contribution of our work lies in extending a base sparse tracker to an efficient dense tracker through a combination of local attention, coarse-to-fine processing, interpolation layers, and upsampling layers. If all these additional components are removed, the remaining base model can indeed perform sparse tracking—this corresponds to the evaluation reported in Table 4 of the paper. However, we would like to emphasize that this evaluation does not reflect the core contribution of our work. For the reviewer’s interest, the explicit steps to downgrade our model for sparse tracking are as follows:
>
> * (i) treat each queried point as an independent token;
> * (ii) disable the interpolator and token activation/densification (i.e., no coarse-to-fine growth); and
> * (iii) disable the final dense upsampler.
>
> We then run the standard k iterative refinements and output the resulting trajectories.
>
> **Question R1.5: Have you tried using other depth estimation models, such as UniDepthv2? It would be interesting to see how a more accurate depth estimator could improve the overall performance.**
>
> Yes. Our main results use UniDepthv2 (ViT-L). In the rebuttal, we replaced depth with a state-of-the-art video estimator (Pi3) and observed consistent gains on 3D tracking metrics, indicating that higher-quality. Please see the answer 2 of our response to common issues.

---

### Author Response · Authors · 2025-11-21
**The General Response to Reviewers (1/4)**

We would like to thank all reviewers for their thoughtful feedback. We are encouraged that several reviewers noted: (i) the paper is clearly written and easy to follow (mKiS, Pxyp); (ii) the method is well designed with thorough analysis (pkxP, ZpXS, mKiS); and (iii) the experiments are extensive and justify the design choices (all reviewers).

**In the updated version of the paper, changes are marked in blue.** The changes to the paper are summarized as follows:
1. Ablation on the number of interpolation neighbors (See Tab. 6).
3. Comparison of 4D correlation's latency on different devices. (see Tab. 7).
4. Ablation on different tracking baseline (See Tab. 9).
5. Comparison with recent sparse 2D/3D tracking methods (See Tab. 11 and Tab. 12 in the Appendix).

---

> ### Author Response · Authors · 2025-11-21
> **The General Response to Reviewers (2/4)**
>
> # Common issues
> Below, we address common questions raised by the reviewers.
>
> **Question C1. About the technical novelty.**
> The reviewers note that the coarse-to-fine strategy and interpolation are not new ideas, and we agree to some extent. Coarse-to-fine is a general and abstract idea that is employed in countless papers, but it is not a drop-in recipe: across vision, many work propose new coarse-to-fine frameworks tailored to their setting—thus crafting an effective variant is itself a substantive contribution.
>
> Our contribution is **the first rigorously analyzed, token-level coarse-to-fine framework for dense, long-term, transformer-based 3D tracking**. It is a non-trivial effort, as we pushed the accuracy–efficiency trade-off of the abstract coarse-to-fine idea very successfully, achieving over 5× speedup with minimal accuracy loss compared to the fastest existing dense tracker, and 19–100× faster runtime than other sparse trackers while improving accuracy in dense tracking settings.
>
> **Why this is non-trivial for 3D tracking:**
> 1. **3D tracking is inherently unstructured.** Although we operate on a regular grid of 3D tracks, this grid only defines neighborhood relationships at the initial frame. As time progresses, neighboring tracks in the original grid may diverge significantly, making consistent coarse-to-fine refinement nontrivial.
> 2. **Recent trends in tracking favor single-resolution models**. Modern tracking frameworks avoid coarse-to-fine to reduce error accumulation. Designing a coarse-to-fine strategy that maintains high accuracy requires careful trade-offs and is widely recognized as a difficult problem in tracking.
> 3. **Coarse-to-fine designs remain underexplored in transformer-based trackers.** Many state-of-the-art systems—e.g., FlowFormer [1], TransFlow [2], CoTracker [3,4], SpatialTracker [5], and DELTA[5]—operate at a single token density rather than adopting a token-level coarse-to-fine hierarchy. While a few transformer variants attempt multi-scale processing, the design space is still limited, and we view a principled, token-level scheme as a meaningful research contribution.
>
> **On related paradigms:**
> 1. **Optical flow**: while earlier optical-flow pipelines (FlowNet[6], PWCNet[7]) have coarse-to-fine designs, tracking is fundamentally different. 3D tracking is more difficult, because particles have to be tracked over a longer time. Optical flow in some sense always resets between two frames and doesn't have these long term temporal dependencies. The insight that a coarse-to-fine approach can work for 2D and 3D tracking alone is an important novel insight.
>
> 2. **DOT[8] (two-stage dense tracking)**. DOT’s coarse (sparse tracking) and fine (optical flow) stages are **separate, heterogeneous modules**. Our approach keeps everything **inside one transformer pipeline**, using a **lightweight, learnable interpolator** to activate new tokens between refinement steps.
>
> 3. **DELTA (frame-level “coarse”)**. DELTA downsamples frames once but still processes **every low-res token in every iteration**, leaving per-iteration attention cost unchanged. Our method is token-level: start very sparse (4–8×), interpolate dense proposals, progressively densify, then finish with one full pass for pixel-level accuracy.
>
> Different from these approaches, our work poses an orthogonal question: can we reduce the token load within the transformer while preserving pixel-level accuracy? We address this by introducing a token-level sparsity schedule that begins with extremely sparse sampling (stride 4×–8×), interpolates dense proposals using a learnable, feature-aware interpolator, and progressively densifies the token set over subsequent iterations. The full design space is detailed in Section 3.2. Below, we briefly summarize our key findings:
>
> 1. **Which dimension to downsample?** Spatial, temporal, and trajectory subsampling are benchmarked in Table 1; only trajectory subsampling preserves accuracy.
> 2. **Which schedule?** Figure 5a and 5c show runtime vs. accuracy trade-offs across different sparsity schedules. A very-sparse-to-dense schedule achieves the best balance.
> 3. **Which sampling pattern?** Table 5 compares uniform grid, random, and SIFT-based patterns. The uniform grid proves to be the most reliable.
> 4. **Which interpolator?** Figure 5b and Supplementary Figures 4–7 evaluate nearest, bilinear, and our attention-based interpolator. The learnable interpolator consistently performs best, especially at large strides.
>
> With these design choices, our final model matches DELTA’s accuracy while achieving a 5× speedup. Importantly, our scheduling strategy is model-agnostic: when combining DELTA’s upsampler and our coarse-to-fine pipeline with CoTracker3, we observe a 3× runtime gain. This demonstrates that our contribution is not tied to a specific architecture.

---

> ### Author Response · Authors · 2025-11-21
> **The General Response to Reviewers (3/4)**
>
> **Question C2. About comparison with recent sparse 2D/3D tracking methods**
>
> First, we believe that evaluation on sparse benchmarks primarily reflects the performance of the **base model**, which is not the focus of our contribution. The core components of our work—the **coarse-to-fine framework** and **interpolation layers**—were not utilized in the following evaluation.
>
> We add a 3D evaluation on TAP-Vid3D against strong sparse trackers—TrackOn2 [9], TAPNext [10], CoTracker3 [4]—and the sparse 3D tracker TAPIP3D [11]. For a fair comparison, all methods are lifted to 3D with the same depth backbone (UniDepth v2, ViT-L); the native 3D methods (SpaTracker, TAPIP3D, and ours) also use UniDepth. This setup matches the protocol in Tab. 4 of our paper. We additionally report APD$\_{3D}^{all}$—the percentage of points whose 3D error falls within a threshold over **both visible and occluded** points—to assess how well models predict occluded trajectories. This complements APD$\_{3D}$, which is computed only on visible points. We report the average results across 3 subsets of TAPVid3D in Table C.1 here (the detailed results are reported in Tab.8 and Tab.9 in our supplementary):
>
> **Table C.1** Average 3D Tracking Results on TAP-Vid3D, using UniDepthV2. $^{\dagger}$ denotes using depth to lift 2D tracks to 3D tracks.
> | Method     | Avg AJ$\uparrow$  | Avg APD$_{3D}^{vis}\uparrow$ | Avg APD$_{3D}^{all}\uparrow$ | Avg OA$\uparrow$ |
> |-------------|:--------:|:------------------------------------:|:-------------------------------------:|:--------:|
> | TrackOn2†   | 12.5     | 19.5                                 | 15.2                                  | **88.2**     |
> | TAPNext†    | 12.5     | 19.7                                 | 15.5                                  | 80.4     |
> | CoTracker3† | 12.3     | 19.5                                 | 15.4                                  | 83.9     |
> | SpaTracker  | 10.0     | 16.8                                 | 13.4                                     | 83.0     |
> | TAPIP3D     | 12.4     | 19.7                                 | 16.2                                  | 84.6     |
> | **Our base tracker**    | **13.4** | **21.0**                             | **17.8**                              | 81.2     |
>
> Across TAP-Vid3D, our method consistently surpasses recent strong 2D sparse trackers on 3D point-tracking metrics (APD$\_{3D}$ and APD$\_{3D}^{all}$). Our *Occlusion Accuracy* is lower than TrackOn2 and TAPIP3D, likely because we supervise *both* visible and occluded trajectories, whereas other methods supervise only visible ones (a similar effect appears for CoTracker2 in Tab. 4). Overall, stronger 2D tracker does not guarantee stronger 3D tracking once depth, occlusion, and long-range consistency are considered; moreover, “2D tracking + depth” recovers only visible motion (yielding much lower APD$_{3D}^{all}$), limiting downstream uses (e.g., 3D/4D reconstruction) that require trajectories through occlusions and out-of-frame segments. These findings underscore the need for end-to-end 3D tracking models.
>
> We also evaluate with a stronger, temporally consistent video-depth backbone, Pi3 [12] in Table C.2. Specifically, we replace UniDepth with Pi3 to lift tracks to 3D and re-run the same TAP-Vid3D protocol for all methods above, so any gains reflect depth quality rather than changes in tracking.
>
> **Table C.2** Average 3D Tracking Results on TAP-Vid3D, using Pi3.  $^{\dagger}$ denotes using depth to lift 2D tracks to 3D tracks.
> | Method     | Avg AJ$\uparrow$  | Avg APD$_{3D}^{vis}\uparrow$ | Avg APD$_{3D}^{all}\uparrow$ | Avg OA$\uparrow$ |
> |-------------|:--------:|:------------------------------------:|:-------------------------------------:|:--------:|
> | TrackOn2†   | 22.4     | 32.1                                 | 24.0                                  | 88.1     |
> | TAPNext†    | 22.1     | 32.1                                 | 24.4                                  | 80.4     |
> | CoTracker3† | 21.9     | 31.7                                 | 24.5                                  | 83.9     |
> | SpaTracker  |  13.4    | 21.3             | 17.2     | 84.0
> | TAPIP3D     | **23.7** | **33.8**                             | **27.9**                              | **86.1** |
> | **Our base tracker**    | 23.4     | 33.2                                 | 27.6                                  | 84.8     |
>
> With more temporally consistent video depth, all methods improve, confirming that 3D tracking benefits from advances in visual geometry. In this setting, we are slightly below TAPIP3D. Importantly, our goal is not to propose a new 3D tracker but to remove the dominant bottlenecks in dense tracking: our token-level coarse-to-fine schedule substantially accelerates inference without sacrificing accuracy, and it transfers cleanly to both dense and sparse settings.

---

> > ### Author Response · Authors · 2025-11-22
> > **The General Response to Reviewers (4/4)**
> >
> > References:
> >
> > [1] Huang, Zhaoyang et al. “FlowFormer: A Transformer Architecture for Optical Flow.” European Conference on Computer Vision (2022).
> >
> > [2] Lu, Yawen et al. “TransFlow: Transformer as Flow Learner.” 2023 IEEE/CVF Conference on Computer Vision and Pattern Recognition (CVPR) (2023): 18063-18073.
> >
> > [3] Karaev, Nikita et al. “CoTracker: It is Better to Track Together.” ArXiv abs/2307.07635 (2023).
> >
> > [4] Karaev, Nikita et al. “CoTracker3: Simpler and Better Point Tracking by Pseudo-Labelling Real Videos.” ArXiv abs/2410.11831 (2024).
> >
> > [5] Ngo, T.D. et al. “DELTA: Dense Efficient Long-range 3D Tracking for any video.” ArXiv abs/2410.24211 (2024).
> >
> > [6] Dosovitskiy, Alexey et al. “FlowNet: Learning Optical Flow with Convolutional Networks.” 2015 IEEE International Conference on Computer Vision (ICCV) (2015): 2758-2766.
> >
> > [7] Sun, Deqing et al. “PWC-Net: CNNs for Optical Flow Using Pyramid, Warping, and Cost Volume.” 2018 IEEE/CVF Conference on Computer Vision and Pattern Recognition (2017): 8934-8943.
> >
> > [8] Moing, Guillaume Le et al. “Dense Optical Tracking: Connecting the Dots.” 2024 IEEE/CVF Conference on Computer Vision and Pattern Recognition (CVPR) (2023): 19187-19197.
> >
> > [9] Aydemir, Görkay et al. “Track-On: Transformer-based Online Point Tracking with Memory.” ArXiv abs/2501.18487 (2025).
> >
> > [10] Zholus, Artem et al. “TAPNext: Tracking Any Point (TAP) as Next Token Prediction.” ArXiv abs/2504.05579 (2025).
> >
> > [11] Zhang, Bowei et al. “TAPIP3D: Tracking Any Point in Persistent 3D Geometry.” ArXiv abs/2504.14717 (2025).
> >
> > [12] Wang, Yifan et al. “$\pi^3$: Permutation-Equivariant Visual Geometry Learning.” (2025).

---

### Author Response · Authors · 2025-12-01
**Final Remark by Authors**

We thank the reviewers for their constructive feedback. Below is a brief summary of key concerns and how we addressed them in the rebuttal:

* **Novelty**. While coarse-to-fine is common in computer vision, applying it **inside an iterative tracker** for dense 3D tracking is **under-explored** and **not a drop-in recipe**. Dense long-range tracking is a challenging task (compared to e.g., optical flow) thus requires a careful design to **speed up without sacrificing accuracy**. We clarified how our proposed token-level coarse-to-fine approach differs from related methods, and demonstrated it **generalizability** to other trackers (CoTracker3, SpatialTracker).

* **Motivation**. We explained why progressive dense tracking is needed (rather than (i) sparse matching → dense upsampling, (ii) using sparse trackers to produce dense outputs, or (iii) simple dense tracking as in DELTA), and highlighted the technical impact of our approach for dense tracking pipelines.

* **Additional evaluation**. We added: (i) 3D point-tracking results of recent strong 2D trackers (TrackOn2, TAPNext, CoTracker3) and 3D tracker (TAPIP3D) under different depth models, including UniDepth-v2 and Pi3; (ii) an ablation on the number of interpolation neighbors; (iii) a coarse-to-fine variant of SpatialTracker; and (iv) a runtime study of the 4D-correlation block on differnet GPU types.

Overall, we believe the proposed approach, together with a principled design and ablation study, is practically impactful and constitutes a meaningful, generalizable advance for the tracking field.

---

### Meta-Review · Area_Chair_xR9r · 2026-01-09

**Summary:**

This paper proposes an efficiency-oriented 3D point tracking framework based on a coarse-to-fine densification strategy with attention-based interpolation, together with an accelerated 4D correlation implementation. The authors clearly identify runtime bottlenecks in existing dense tracking pipelines and present a carefully engineered solution that yields meaningful speed improvements while largely preserving baseline accuracy. The paper is generally well written, and the experimental section includes detailed runtime analyses, ablation studies, and speed–accuracy trade-off evaluations, reflecting substantial engineering effort.

Initially, the evaluation, while thorough in some aspects, also had notable limitations. The accelerated 4D correlation was only benchmarked on a single GPU architecture (A100), leaving questions about generality across hardware. Similarly, the proposed acceleration strategy was not validated across multiple 3D tracking architectures, despite being presented as broadly applicable. Comparisons with recent strong baselines—particularly 2D trackers with depth lifting such as CoTracker3, Track-On, and TAPNext—were missing in key benchmarks, making it difficult to fully contextualize the reported gains. However, these experimental limitations were well addressed during rebuttal.

Despite these strengths, as raised by reviewers, AC do not find the research contribution sufficiently novel or conceptually distinct to support acceptance.

The core ideas—progressive densification, coarse-to-fine refinement, and local interpolation-based updates—are well established in prior literature, including dense matching and optical-flow pipelines. The proposed learnable interpolator closely resembles update mechanisms used in DELTA and TAPTRv2, and the overall coarse-to-fine design mirrors refinement strategies explored in optical flow models such as FlowFormer, albeit applied at the token or trajectory level. As a result, the method reads primarily as an engineering-driven extension of existing frameworks, rather than a new tracking paradigm or representation of motion and geometry.

Moreover, the paper does not sufficiently justify why progressive dense association is fundamentally preferable to alternative sparse or correlation-based tracking strategies beyond empirical runtime gains. The motivation remains largely practical rather than conceptual, and the work does not introduce new insights into motion modeling, geometric reasoning, or learning dynamics that would meaningfully advance the field.

Finally, several design choices remain insufficiently analyzed. The impact of the fixed number of nearest neighbors in the interpolator is not explored, and some experimental results (e.g., learnable interpolator underperforming simpler schemes, performance degradation with increasing iterations) are not clearly explained or discussed. Additionally, although efficiency is emphasized as a core motivation, the reported memory footprint remains very large, weakening the overall efficiency narrative.

In summary, while the paper demonstrates solid engineering, careful experimentation, and practical speed improvements, its conceptual novelty and research insight are limited, and the contribution is best viewed as an incremental efficiency enhancement of prior methods rather than a standalone research advance. For these reasons, AC recommend rejection.

**Reviewer Concerns:**

Initially, the evaluation, while thorough in some aspects, also had notable limitations. The accelerated 4D correlation was only benchmarked on a single GPU architecture (A100), leaving questions about generality across hardware. Similarly, the proposed acceleration strategy was not validated across multiple 3D tracking architectures, despite being presented as broadly applicable. Comparisons with recent strong baselines—particularly 2D trackers with depth lifting such as CoTracker3, Track-On, and TAPNext—were missing in key benchmarks, making it difficult to fully contextualize the reported gains. However, these experimental limitations were well addressed during rebuttal.

**Reviewer Scores:**

They might not be changed.

---

### Decision · Program_Chairs · 2026-01-26

Reject